# ROUTER CHOICE MATTERS: RANK-AWARE POST-TRAINING QUANTIZATION FOR MoE MODELS

## ABSTRACT

Quantizing Mixture-of-Experts (MoE) language models is challenging since router errors cascade into expert selection and dominate accuracy loss. We study this effect and show that preserving router decisions of the selected experts yields the largest gains, with most errors arising as near-neighbor rank flips around the top-$k$ experts. Motivated by these observations, we present ExpertQuant, a training-free, calibration-only post-training quantization (PTQ) framework tailored to MoE. ExpertQuant combines (i) *Expert-Aware Scale* to accommodate heterogeneous activation ranges and two router-alignment objectives between quantized and full-precision models: (ii) *Rank-Aware Jaccard Loss*, which aligns the top-$k$ expert rank, and (iii) *Gap Hinge Loss*, which preserves score margins between consecutive experts to suppress rank flipping. Across OLMoE, DeepSeek-MoE, and Qwen3-MoE, ExpertQuant consistently reduces perplexity on C4 and WikiText-2 and improves zero-shot accuracy under W4A4 and W4A8, with similar trends at lower bit-widths. The framework requires no retraining, integrates seamlessly with existing MoE, and demonstrates that stabilizing router rankings during calibration is key to accurate low-bit MoE inference.

## 1 INTRODUCTION

Large language models (LLMs) continue to advance rapidly and reshape modern natural language processing (Achiam et al., 2023; Grattafiori et al., 2024; Guo et al., 2025; Yang et al., 2025). As parameter counts and training corpora grow, Mixture-of-Experts (MoE) architectures emerge as a scalable design that raises effective capacity without proportional compute (Shazeer et al., 2017; Fedus et al., 2022; Dai et al., 2024b; Muennighoff et al., 2025). An MoE layer comprises a learned router and a pool of experts; for each input token, the router computes routing scores, activates the top-$k$ experts, and aggregates their outputs. Variants include shared experts that capture common knowledge across tokens, while routed experts specialize. As model size and MoE adoption increase, deployment becomes constrained by memory and latency, so low-precision inference via quantization becomes essential for practical serving.

Quantizing MoE models differs fundamentally from quantizing dense transformers because the router determines which experts are activated. When the router selects suboptimal experts, the entire forward pass is affected, leading to accuracy loss. This makes router performance more critical in MoE than in dense architectures. To validate this, we conduct controlled studies in which a module is kept in full-precision and others are quantized. Importantly, across all settings, the experts' FFNs are always quantized. As shown in Figure 1, preserving router performance consistently yields the highest performance, confirming that router accuracy is the dominant factor in MoE quantization. We further analyze router errors and find that, after

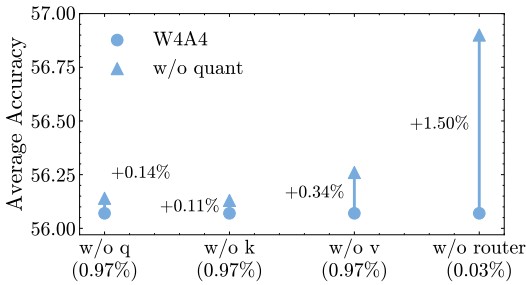

Figure 1: Controlled study on W4A4 OLMoE, where w/○ denotes the unquantized module. The percentages on the $x$-axis indicate the proportion of the whole parameters.

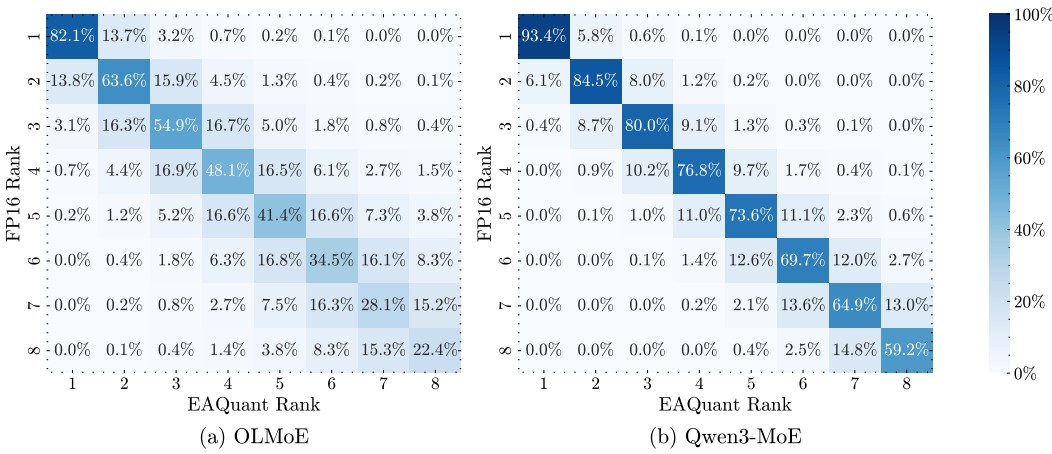

Figure 2: Confusion matrices at **layer 0** comparing FP16 vs. EAQuant top-$k$ indices under W4A4.

quantization, the router typically selects experts that are close in rank to the full-precision choices, with mistakes concentrated near the top-$k$ experts rather than spread arbitrarily. This is evident in the confusion matrix of Figure 2[1], where errors cluster near the diagonal, showing near-neighbor expert flips. These findings suggest that effective PTQ of MoE must explicitly stabilize router rankings.

Existing PTQ methods primarily target dense models and do not directly address these MoE-specific issues (Frantar et al., 2022; Lin et al., 2024b; Xiao et al., 2023; Lin et al., 2024a). Although MoE-Quant reweights expert calibration by router weights, it does not handle the quantization-induced router discrepancy (Chen et al., 2025). While EAQuant calibrates the router with KL divergence (Fu et al., 2025), this aligns score distributions but does not preserve the margin between adjacent experts and therefore fails to reduce rank inversions. To close this gap, we introduce two router-alignment objectives that directly compare the quantized and full-precision routers: *Rank-Aware Jaccard Loss (RAJ)*, which aligns their rank of top-$k$ experts, and *Gap Hinge Loss (GH)*, which keeps score margins between adjacent experts to suppress rank flips. In addition, we propose an *Expert-Aware Scale (ES)* that assigns each expert its own channel-wise scale to match per-expert activation and weight ranges, preventing heavy-tailed experts from dominating shared scales. Together, these objectives form an MoE-aware PTQ framework that improves low-bit accuracy.

Our main contributions are as follows:

- We demonstrate that the PTQ accuracy of MoE is primarily determined by router performance, which accounts for the majority of performance degradation.

- We identify a router failure, i.e., near-neighbor rank flips, and propose *Rank-aware Jaccard Loss* and *Gap Hinge Loss* to stabilize expert selection and preserve margins.

- We validate our framework on OLMoE, DeepSeek-MoE, and Qwen3-MoE, achieving lower perplexity on C4 and WikiText-2 and higher accuracy on diverse reasoning tasks, all within a training-free, calibration-only pipeline.

## 2 RELATED WORK

### 2.1 MIXTURE-OF-EXPERTS LARGE LANGUAGE MODELS

Early studies propose using gating networks to adaptively route each input to specialized sub-networks (Jacobs et al., 1991; Jordan & Jacobs, 1994), and subsequent work extends this idea to a variety of domains (Deisenroth & Ng, 2015; Aljundi et al., 2017). In LLMs, an MoE layer places expert MLPs behind a lightweight gate (linear projection plus softmax) and routes each token to the top-$k$ experts with load-balancing regularization (Shazeer et al., 2017); systems advance scale MoE transformers with automated sharding and parallelism (Lepikhin et al., 2021).

---

[1]Confusion matrices for additional OLMoE layers are shown in Figures 9 and 10.

Large-scale instances vary in routing. Switch transformer uses top-1 gating to reduce activation cost (Fedus et al., 2022); GLaM shows that top-2 improves the accuracy, an efficiency trade-off at trillion parameter scale (Du et al., 2022); Mixtral 8×7B activates two experts per token and rivals dense peers at similar cost (Jiang et al., 2024). DeepSeek-MoE increases expert granularity, keeps a few active experts per token, and adds always-on shared experts to capture global knowledge (Dai et al., 2024b); DeepSeek-V2/V3 further refine routing, optimization, and systems (Liu et al., 2024a;b).

Unlike dense transformers, which apply all parameters to every token, MoE models rely on a router to decide which subset of experts is activated. This router is therefore central to both efficiency and accuracy: small perturbations in its outputs directly change expert selection and propagate through the entire forward pass. Our work focuses on this router behavior and its interaction with expert heterogeneity, distinguishing MoE-specific challenges from those in dense architectures.

## 2.2 POST-TRAINING QUANTIZATION FOR LLMs

PTQ is a standard route to deploy LLMs efficiently, reducing memory and bandwidth without re-training. In dense transformers, PTQ minimizes layerwise reconstruction error while preserving numerical structure relevant to generation. GPTQ casts weight-only quantization as a blockwise least-squares problem with Hessian-aware error compensation, delivering strong 4-bit accuracy at negligible calibration cost (Frantar et al., 2022). AWQ accounts for activation statistics during calibration, preserves high-saliency channels, and uses data-aware scaling to control outliers (Lin et al., 2024b). DuQuant redistributes activation outliers via a dual transformation that rebalances ranges in both activation and weight, enabling competitive W4A4 across dense LLMs (Lin et al., 2024a).

In MoE architectures, the router is the most critical component as its outputs determine which experts are activated; even small quantization errors can cascade into misrouted tokens and dominate accuracy loss. Existing MoE-specific PTQ methods only partially address this challenge. Although MoEQuant reweights expert calibration by router weights, it does not directly correct the quantization-induced discrepancy in router decisions (Chen et al., 2025). EAQuant applies an additional KL-divergence objective to align router logit distributions, but this merely matches score distributions and does not explicitly address rank inversions or prevent expert flipping (Fu et al., 2025). In contrast, our approach directly targets router stability: we aim to preserve the ranking of selected experts while also preserving score margins between them, thereby reducing the likelihood of rank flips and improving robustness during inference.

## 3 METHODOLOGY

PTQ of MoE is challenging since experts have heterogeneous statistics, and the router's top-$k$ selection is brittle to small logit noise. We address these issues with three objectives. First, §3.2 introduces an *Expert-Aware Scale* that assigns each expert its own channelwise factor, balancing activation and weight ranges per channel. Second, §3.3 proposes a *Rank-Aware Jaccard Loss* that preserves the ranking of experts between the quantized and full-precision routers. Third, §3.4 introduces a *Gap Hinge Loss* that preserves score margins to stabilize expert ordering. As illustrated in Figure 3, the *Expert-Aware Scale* calibrates per-expert quantization, while the two router losses align selection with FP16, together enabling robust MoE quantization.

### 3.1 PRELIMINARIES

**Mixture of Experts.** An MoE layer comprises a router and a pool of experts. Given a token representation, the router produces routing scores and activates only a small subset of experts (sparse gating), typically the top-$k$ with the highest scores. Each selected expert processes the same input in parallel, and the layer aggregates their outputs with router-derived weights. In language models, experts are usually lightweight position-wise feed-forward networks (FFNs/MLPs), enabling high capacity with limited compute by keeping activation sparse across tokens (Shazeer et al., 2017; Fedus et al., 2022). Formally, the MoE output aggregates selected expert outputs as

$$\mathbf{y} = \sum_{j \in \text{top-}k\big(\mathbf{g}(\mathbf{x})\big)} \pi_j(\mathbf{x})\, \mathcal{E}_j(\mathbf{x}), \tag{1}$$

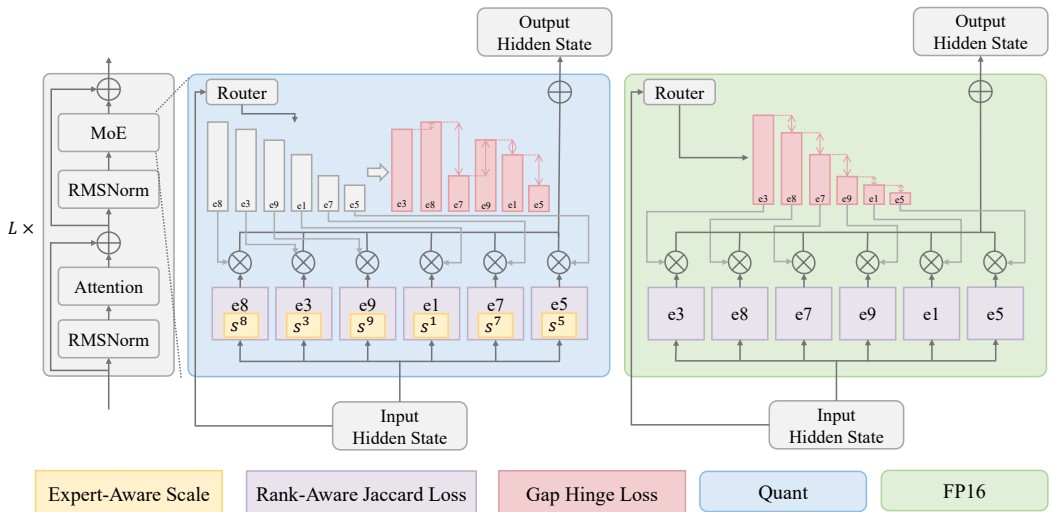

Figure 3: The overview of ExpertQuant.

where the router produces scores via router $\mathbf{g}$, $\text{top-}k$ selects the active experts, $\pi_j(\mathbf{x})$ denotes the normalized routing weight for expert $j$, and $\mathcal{E}_j$ denotes the expert function. This design yields conditional computation that scales model capacity while preserving inference efficiency.

**Post-Training Quantization.** PTQ converts a floating-point network into a low-precision model without gradient updates. Given a small calibration set, PTQ estimates activation ranges, chooses a quantization scheme (e.g., uniform affine, symmetric or asymmetric), and assigns scales at appropriate granularities. In transformer blocks, linear-layer weights typically use per-channel quantization for stronger error control, while activations use per-token or per-tensor scales for amortized overhead. Runtime inserts integer arithmetic on the quantized path and dequantizes only at layer boundaries needed by residual connections or mixed-precision kernels. A generic $b$-bit uniform affine quantizer with scale $s > 0$ and integer zero-point $z$ reconstructs a dequantized tensor $\hat{\mathbf{x}}$ from a real-valued tensor $\mathbf{x}$ as

$$\hat{\mathbf{x}} = s \left( \text{clip}\left( \left\lfloor \frac{\mathbf{x}}{s} \right\rceil + z,\ q_{\min},\ q_{\max} \right) - z \right), \tag{2}$$

where $q_{\min}$ and $q_{\max}$ denote the integer bounds (e.g., $-2^{b-1}$ to $2^{b-1} - 1$ for symmetric $b$-bit), and $\lfloor \cdot \rceil$ is the rounding function. Practical refinements include bias correction, range smoothing for outliers, and rounding optimization to minimize the reconstruction error of salient channels (Jacob et al., 2018; Frantar et al., 2022; Xiao et al., 2023; Lin et al., 2024b).

## 3.2 EXPERT-AWARE SCALE (ES)

Prior work in the multi-expert setting first computes a per-channel scale for each expert and then takes the maximum across experts, which forces all experts in the same layer to share one scale per channel (Fu et al., 2025); this max-aggregation is dominated by a few heavy-tailed experts and therefore either clips light-tailed experts or wastes quantization levels on the majority. This can be observed from the activation landscapes in Figure 4 that experts exhibit markedly different activation and weight statistics.[2] We therefore choose to assign each expert its own channel-wise scale and implement it through an exact diagonal reparameterization. Let an MoE MLP take input $\boldsymbol{x} \in \mathbb{R}^d$, and let expert $i \in \{1, \ldots, E\}$ use weights $W^i \in \mathbb{R}^{m \times d}$ with column $\mathbf{W}_j^i$ for channel $j$. Following the scaling strategy of SmoothQuant (Xiao et al., 2023), we introduce a positive diagonal matrix $D_i = \text{diag}(\boldsymbol{s}^i)$, where $\boldsymbol{s}^i = (s_1^i, \ldots, s_d^i)$, and rewrite the expert matrix multiplication as

$$W^i \boldsymbol{x} = (W^i D_i)(D_i^{-1} \boldsymbol{x}) \triangleq \widetilde{W}^i \widetilde{\boldsymbol{x}}^i, \tag{3}$$

which is algebraically exact in full precision; we then quantize $\widetilde{W}^i$ per channel and $\widetilde{\boldsymbol{x}}^i$ per token. The key is to choose $\boldsymbol{s}^i$ to balance the ranges of the two operands for each expert and channel. Let

---

[2]Expert activations for additional layers for each model can be found from Figures 11 to 14.

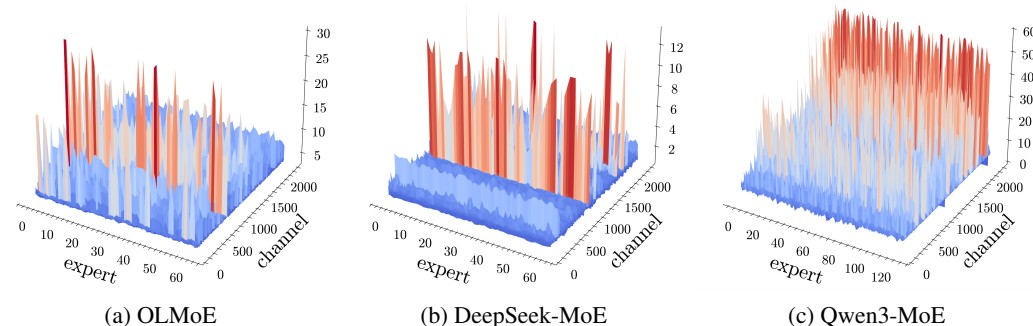

(a) OLMoE        (b) DeepSeek-MoE        (c) Qwen3-MoE

Figure 4: Expert activations in the **last layer** for each model.

$|x_j|$ be the magnitude of the $j$-th input activation observed on a small calibration set, and $|\mathbf{W}_j^i|$, the absolute values of the corresponding weight column. We set the *ES* by

$$s_j^i = \frac{\max(|x_j|)^\alpha}{\max(|\mathbf{W}_j^i|)^{1-\alpha}}, \qquad j = 1, \ldots, d, \ \ i = 1, \ldots, E, \ \ \alpha \in [0, 1], \tag{4}$$

to trade off how much normalization burden is placed on activations versus weights. In practice, the maxima in Equation 4 can be replaced by high-percentile statistics for robustness and $s_j^i$ is clipped to $[s_{\min}, s_{\max}]$ to avoid extreme rescaling; both choices keep the form of Equation 4 unchanged. Compared with the max-aggregation baseline (one shared $s_j$ across experts), the expert-aware design adapts to each expert's local statistics, reduces activation clipping without destabilizing weight quantization, preserves expert outputs that the router relies on, and thus improves MoE quantization with negligible runtime cost (see Table 13).

### 3.3 Rank-Aware Jaccard Loss (RAJ)

As shown in Figure 2, quantization errors in the router are not arbitrary: the quantized router usually selects experts that are close in rank to the FP16 choices, with most mistakes occurring near the top-$k$ experts. This suggests that aligning the set and order of selected experts is more important than matching all experts, including those not selected. In particular, correcting which experts appear in the top-$k$ list, especially at higher ranks, can recover most of the lost accuracy. To capture this phenomenon, we design a rank-weighted similarity objective. Let the FP16 and quantized router logits be $\mathbf{r}^{(\mathrm{fp})}, \mathbf{r}^{(\mathrm{q})} \in \mathbb{R}^E$, and denote their ordered top-$k$ expert indices as

$$I = (i_1, \ldots, i_k) = \text{top-}k(\mathbf{r}^{(\mathrm{fp})}), \quad J = (j_1, \ldots, j_k) = \text{top-}k(\mathbf{r}^{(\mathrm{q})}),$$

where $I$ and $J$ are the ordered sets of the selected expert indices for the full-precision and quantized models, respectively. We assign geometric weights $w_r = \beta^{r-1}$ with $\beta \in (0, 1]$ so that higher ranks receive larger emphasis. Using these, we define affinity vectors $A_{\mathrm{fp}}, A_{\mathrm{q}} \in \mathbb{R}^E$:

$$A_{\mathrm{fp}}(e) = \begin{cases} w_r, & e = i_r, \\ 0, & \text{otherwise}, \end{cases} \qquad A_{\mathrm{q}}(e) = \begin{cases} w_r, & e = j_r, \\ 0, & \text{otherwise}. \end{cases}$$

The *RAJ* similarity is then

$$J_W(I, J) = \frac{\sum_{e=1}^E \min(A_{\mathrm{fp}}(e), A_{\mathrm{q}}(e))}{\sum_{e=1}^E \max(A_{\mathrm{fp}}(e), A_{\mathrm{q}}(e))} \in [0, 1], \tag{5}$$

and the corresponding loss is $\mathcal{L}_{\mathrm{RAJ}} = 1 - J_W(I, J)$.

This design directly measures the agreement between FP16 and quantized top-$k$ experts while prioritizing higher ranks. Unlike logit-based losses, it is invariant to affine transformations of $\mathbf{r}$ that preserve ordering, and becomes lower whenever a near-miss occurs (e.g., swapping rank-$k$ and rank-$(k+1)$ experts). In this way, $\mathcal{L}_{\mathrm{RAJ}}$ focuses optimization exactly where MoE routers are most brittle around the top-$k$ experts.

### 3.4 GAP HINGE LOSS (GH)

While *RAJ* aligns the top-$k$ expert order, it does not control how close the router scores are between adjacent experts. If two experts have nearly equal scores, even small quantization noise can flip their ranks and destabilize routing. To address this, we explicitly encourage larger gaps between consecutive experts, making the selection more robust to perturbations.

Formally, we extract paired FP16 and quantized scores on the fixed FP16 top-$k$ index set:

$$r_r^{(\text{fp})} = \mathbf{r}_{i_r}^{(\text{fp})}, \qquad \tilde{r}_r^{(\text{q})} = \mathbf{r}_{i_r}^{(\text{q})}, \quad r = 1, \dots, k, \tag{6}$$

and define consecutive FP16 and quantized gaps as

$$\Delta_r^{(\text{fp})} = r_r^{(\text{fp})} - r_{r+1}^{(\text{fp})}, \qquad \Delta_r^{(\text{q})} = \tilde{r}_r^{(\text{q})} - \tilde{r}_{r+1}^{(\text{q})}, \quad r = 1, \dots, k-1. \tag{7}$$

The *GH* loss penalizes quantized gaps that shrink below the FP16 reference (up to $\gamma \geq 0$):

$$\mathcal{L}_{\text{GH}} = \frac{1}{k-1} \sum_{r=1}^{k-1} \left[ \Delta_r^{(\text{fp})} - \Delta_r^{(\text{q})} + \gamma \right]_+, \qquad \text{where } [x]_+ = \max(0, x). \tag{8}$$

Since our target is to preserve the margins, we simply set $\gamma$ to 0. By preserving the logit gaps, $\mathcal{L}_{\text{GH}}$ reduces the probability of rank inversions caused by quantization.

Finally, we combine *RAJ* and *GH* into a unified router-consistency objective:

$$\mathcal{L}_{\text{router}} = \lambda_{\text{RAJ}} \mathcal{L}_{\text{RAJ}} + \lambda_{\text{GH}} \mathcal{L}_{\text{GH}}, \qquad \lambda_{\text{RAJ}}, \lambda_{\text{GH}} \geq 0, \tag{9}$$

where *RAJ* pulls the correct FP16 experts into the quantized top-$k$, and *GH* preserves the gaps that keep them there. Together, they jointly mitigate routing errors induced by quantization. We also provide a theoretical analysis in Section E explaining how *RAJ* and *GH* improve router stability from the perspective of output differences.

## 4 EXPERIMENTS

### 4.1 SETTINGS

We conduct all experiments on a single NVIDIA A100 (80 GB) GPU using `PyTorch` and fix the random seed across runs. For post-training calibration, we sample 256 sequences from the C4 corpus with a sequence length of 2048 tokens (Raffel et al., 2020). We use DuQuant (Lin et al., 2024a) as our underlying quantization framework and apply per-token activation quantization and per-channel weight quantization, following prior work (Fu et al., 2025). We configure the router with 8-bit weights and 8-bit activations (W8A8) in all experiments and ablation studies unless otherwise specified. The hyperparameters are set to $\alpha = 0.6$, $\beta = 0.95$, $\lambda_{\text{RAJ}} = 1$ and $\lambda_{\text{GH}} = 1$ throughout all experiments. We evaluate our method on state-of-the-art MoE models: OLMoE (Muennighoff et al., 2025), DeepSeek-MoE (Dai et al., 2024a), and Qwen3-MoE (Yang et al., 2025).

### 4.2 DATASET

We evaluate perplexity on WikiText-2 (Merity et al., 2017) and C4 (Raffel et al., 2020). For zero-shot accuracy, we use ARC-Challenge, ARC-Easy (Clark et al., 2018), BoolQ (Clark et al., 2019), HellaSwag (Zellers et al., 2019), OpenBookQA (Mihaylov et al., 2018), RTE (Dagan et al., 2005), and WinoGrande (Sakaguchi et al., 2021). We follow the standard zero-shot protocol (no in-context examples) and score multiple-choice options by their average token log-likelihood.

### 4.3 EVALUATION METRICS

We evaluate models using token-level perplexity, zero-shot accuracy, and router consistency. Perplexity measures the model's average surprise over the evaluation corpus, where lower values indicate better language modeling; accuracy measures the fraction of correct predictions on downstream tasks, where higher values are better. We compute perplexity with each model's native tokenizer and measure accuracy under the standard zero-shot protocol. For zero-shot benchmarks, we use the open-source `lm-evaluation-harness` (v0.4.9.1) to standardize prompting and scoring under its default configuration (Gao et al., 2024). Router consistency is quantified by the *Match Score* defined in Appendix A, and complete per-task results appear in Appendix B.

Table 1: **Main results under W4A4.** DuQuant serves as the baseline; the comparison includes MoEQuant, EAQuant, and ExpertQuant on OLMoE, DeepSeek-MoE, and Qwen3-MoE.

| Model | Method | Perplexity ↓ | | Accuracy ↑ | | | | | | | |
|-------|--------|------|------|-------|-------|-------|--------|-------|-------|-------|---------|
| | | Wiki2 | C4 | ARC-C | ARC-E | BoolQ | HellaS | OBQA | RTE | Wino. | Average |
| OLMoE | FP16 | 6.65 | 10.86 | 46.59 | 77.10 | 70.09 | 58.47 | 32.60 | 71.12 | 68.51 | 60.64 |
| | DuQuant | 8.28 | 12.20 | 40.96 | 72.26 | 63.88 | 54.99 | 30.20 | 60.43 | 64.72 | 55.35 (-) |
| | MoEQuant | 8.03 | 12.02 | 40.70 | 73.44 | 66.30 | 55.69 | 29.60 | 62.09 | 63.38 | 55.89 (+0.97%) |
| | EAQuant | 7.75 | 11.75 | 41.38 | **73.65** | **66.97** | 56.05 | 31.00 | 62.45 | 65.43 | 56.70 (+2.45%) |
| | ExpertQuant | **7.73** | **11.73** | **43.17** | 72.85 | 65.57 | **56.21** | 32.00 | 69.31 | 65.67 | **57.83 (+4.48%)** |
| DeepSeek | FP16 | 6.51 | 9.10 | 45.14 | 75.88 | 72.69 | 58.10 | 32.40 | 62.82 | 70.32 | 59.62 |
| | DuQuant | 7.66 | 10.52 | 39.25 | 71.33 | 64.89 | 53.54 | 25.00 | 57.04 | 64.64 | 53.67 (-) |
| | MoEQuant | 7.55 | 10.24 | 39.08 | 70.83 | 66.67 | 53.54 | 26.60 | 58.24 | 64.09 | 54.15 (+0.89%) |
| | EAQuant | 7.34 | 10.07 | 38.40 | 71.00 | 68.44 | 54.68 | 26.80 | 57.04 | **65.82** | 54.60 (+1.73%) |
| | ExpertQuant | **7.31** | **10.06** | 40.19 | 72.01 | 68.65 | 55.12 | 28.80 | 59.93 | 64.40 | **55.59 (+3.57%)** |
| Qwen3 | FP16 | 8.70 | 12.31 | 52.56 | 79.34 | 88.75 | 59.52 | 34.00 | 83.03 | 70.32 | 66.79 |
| | DuQuant | 9.86 | 13.62 | 46.33 | 73.44 | **86.67** | 56.71 | **32.60** | **80.51** | 64.25 | 62.93 (-) |
| | MoEQuant | 9.85 | 13.58 | 47.18 | 73.99 | 86.64 | 55.29 | 31.80 | 80.14 | 65.82 | 62.98 (+0.08%) |
| | EAQuant | 9.59 | 13.29 | 49.15 | 75.29 | 86.48 | **56.72** | 30.40 | 77.62 | 66.46 | 63.16 (+0.37%) |
| | ExpertQuant | **9.58** | **13.28** | **49.66** | **75.84** | 86.64 | 56.69 | 32.40 | 79.78 | 66.61 | **63.95 (+1.61%)** |

Table 2: **Main results under W4A8.** DuQuant serves as the baseline; the comparison includes MoEQuant, EAQuant, and ExpertQuant on OLMoE, DeepSeek-MoE, and Qwen3-MoE.

| Model | Method | Perplexity ↓ | | Accuracy ↑ | | | | | | | |
|-------|--------|------|------|-------|-------|-------|--------|-------|-------|-------|---------|
| | | Wiki2 | C4 | ARC-C | ARC-E | BoolQ | HellaS | OBQA | RTE | Wino. | Average |
| OLMoE | FP16 | 6.65 | 10.86 | 46.59 | 77.10 | 70.09 | 58.47 | 32.60 | 71.12 | 68.51 | 60.64 |
| | DuQuant | 7.30 | 11.41 | 43.60 | 73.53 | 65.93 | 56.98 | 31.20 | 62.09 | 66.54 | 57.12 (-) |
| | MoEQuant | 7.22 | 11.30 | **43.94** | **75.21** | 65.47 | 56.52 | 31.00 | 66.45 | 66.85 | 57.92 (+1.39%) |
| | EAQuant | 7.27 | **11.25** | 41.78 | 73.24 | 67.22 | 56.52 | **31.80** | 67.51 | 66.92 | 57.86 (+1.28%) |
| | ExpertQuant | **7.14** | **11.25** | 41.89 | 72.85 | **67.68** | **57.16** | 31.40 | **68.59** | 67.09 | **58.09 (+1.70%)** |
| DeepSeek | FP16 | 6.51 | 9.10 | 45.14 | 75.88 | 72.69 | 58.10 | 32.40 | 62.82 | 70.32 | 59.62 |
| | DuQuant | 7.02 | 9.70 | 41.47 | 73.44 | 70.37 | 56.21 | **29.20** | 60.28 | 67.32 | 56.90 (-) |
| | MoEQuant | 6.90 | 9.58 | 41.47 | 73.19 | 72.08 | **56.75** | 28.60 | 59.21 | 67.96 | 57.04 (+0.24%) |
| | EAQuant | 6.89 | 9.58 | 41.89 | 74.07 | 71.38 | 56.34 | **29.20** | 60.29 | 67.96 | 57.30 (+0.71%) |
| | ExpertQuant | **6.88** | **9.56** | 42.83 | 74.28 | 73.12 | 56.68 | **29.20** | 61.37 | 68.27 | **57.96 (+1.87%)** |
| Qwen3 | FP16 | 8.70 | 12.31 | 52.56 | 79.34 | 88.75 | 59.52 | 34.00 | 83.03 | 70.32 | 66.79 |
| | DuQuant | 9.20 | 12.69 | 52.13 | **79.42** | 86.35 | 58.07 | 33.40 | 79.03 | 66.98 | 65.05 (-) |
| | MoEQuant | 9.19 | 12.67 | 51.28 | 77.78 | **88.87** | 58.17 | 33.80 | 80.51 | 67.88 | 65.47 (+0.64%) |
| | EAQuant | 9.13 | 12.62 | 51.54 | 78.20 | 85.85 | 58.52 | 35.00 | 80.87 | 67.32 | 65.33 (+0.42%) |
| | ExpertQuant | **9.01** | **12.61** | 52.99 | 78.54 | 87.95 | **58.53** | 35.40 | 82.67 | 69.14 | **66.46 (+2.16%)** |

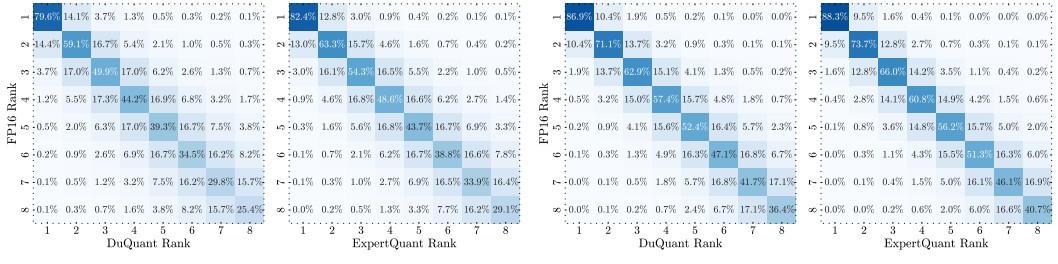

(a) W4A4      (b) W4A8

Figure 5: Layer-averaged confusion matrices of router top-$k$ indices versus FP16 on OLMoE; ExpertQuant shows stronger on-diagonal alignment than DuQuant.

## 4.4 RESULTS

**Main Results.** Tables 1 and 2 summarize PTQ performance on OLMoE, DeepSeek-MoE, and Qwen3-MoE. Across both bit settings, ExpertQuant attains the best average accuracy among PTQ baselines and closes the perplexity gap to full precision. Relative to DuQuant, ExpertQuant improves the average zero-shot accuracy at W4A4 by +4.48% on OLMoE, +3.57% on DeepSeek-MoE, and

Table 4: Ablation of ExpertQuant objectives on OLMoE (W4A4).

| ES | RAJ | GH | ARC-C | ARC-E | BoolQ | HellaS | OBQA | RTE | Wino. | Avg. ↑ |
|----|-----|----|-------|-------|-------|--------|------|-----|-------|--------|
|    |     |    | 40.96 | 72.26 | 63.88 | 54.99  | 30.20 | 60.43 | 64.72 | 55.35 |
| ✓  |     |    | 41.30 | **73.36** | **65.96** | **56.40** | 31.00 | 64.62 | **66.06** | 56.96 (+2.91%) |
| ✓  | ✓   |    | 41.89 | 72.84 | 65.38 | 56.25  | 31.00 | 67.51 | 65.87 | 57.25 (+3.43%) |
| ✓  |     | ✓  | 43.09 | 73.32 | 65.42 | 56.15  | 31.60 | 68.23 | 65.52 | 57.62 (+4.10%) |
| ✓  | ✓   | ✓  | **43.17** | 72.85 | 65.57 | 56.21  | **32.00** | **69.31** | 65.67 | **57.83 (+4.48%)** |

+1.61% on Qwen3-MoE, and at W4A8 by +1.70% on OLMoE, +1.87% on DeepSeek-MoE, and +2.16% on Qwen3-MoE. It also yields the lowest perplexity among quantized methods under W4A4 and W4A8. Overall, ExpertQuant consistently strengthens accuracy while preserving perplexity across diverse MoE backbones and bit budgets, supporting our design of expert-specific scaling and router-consistency losses for robust PTQ of MoE. We also provide the weight-only quantization comparison in Section C.

**Router Consistency (Match Score).** Table 3 reports the agreement between FP16 routing and each PTQ method, while the layer-averaged confusion matrices in Figure 5 visualize the same trend. Across OLMoE, DeepSeek-MoE, and Qwen3-MoE, ExpertQuant achieves the highest match scores under both W4A4 and W4A8, indicating that its quantized router decisions remain closest to the full-precision baseline. Notably, these improvements in routing consistency directly parallel the accuracy gains reported in Tables 1 and 2: methods that better preserve router behavior deliver stronger accuracy at the same bit level. This reinforces our design principle

Table 3: Match Score under W4A4 and W4A8 across different methods.

| Bits | Method | Match Score ↑ | | |
|------|--------|-------|----------|-------|
|      |        | OLMoE | DeepSeek | Qwen3 |
| W4A4 | DuQuant | 63.71 | 56.11 | 51.68 |
|      | MoEQuant | 64.25 | 57.38 | 52.23 |
|      | EAQuant | 65.02 | 58.11 | 52.45 |
|      | ExpertQuant | **66.95** | **59.24** | **53.97** |
| W4A8 | DuQuant | 72.87 | 62.56 | 66.17 |
|      | MoEQuant | 73.69 | 62.88 | 66.97 |
|      | EAQuant | 73.54 | 63.96 | 66.86 |
|      | ExpertQuant | **75.25** | **65.00** | **68.13** |

that stabilizing router rankings is essential for low-bit MoE inference and translates into downstream performance improvements.

## 4.5 ABLATION STUDY

**Objective-wise Impact.** We quantify the contribution of each objective on OLMoE at W4A4 (Table 4). Starting from the DuQuant baseline, *ES* alone raises the average to 56.96. On top of *ES*, *RAJ* lifts performance to 57.25, and *GH* to 57.62, with *GH* providing the larger incremental gain. Enabling all three objectives yields the best result, 57.83. The monotonic gains from *ES* → *ES+RAJ/GH* → *ES+RAJ+GH* align with the match-score trends in Table 3, supporting our design that directly targets router consistency.

**Lower-Precision Robustness.** Pushing weights to 3-bit while keeping activations at 8-bit stresses the MoE pipeline, yet ExpertQuant remains the most robust. All methods experience a drop in accuracy relative to FP16, yet ExpertQuant shows the smallest gap and attains the best average accuracy (Table 5). The trend aligns with our routing hypothesis: lower weight precision amplifies rank flips and narrows score margins, making router consistency increasingly critical. *ES* stabilizes per-expert activation ranges, and *RAJ/GH* improve the match score by preserving rankings and margin gaps, which in turn sustains downstream accuracy under the more aggressive W3A8 setting.

**Rank-Decay Sensitivity** ($\beta$). We study the weighting parameter $\beta$ in *RAJ* (Figure 6), which controls how importance decays across expert ranks: $\beta = 1$ assigns uniform weight to all top-$k$ ranks, while a smaller $\beta$ applies an exponential decay that emphasizes higher-ranked experts. Sweeping $\beta \in \{0.85, 0.90, 0.95, 1.00\}$, we find that $\beta = 0.95$ consistently yields the best average accuracy under both W4A4 and W4A8. $\beta \leq 0.90$ overemphasize the top-1 and top-2 experts and amplify noise from

Table 5: W3A8 results on the OLMoE.

| Method | Wiki2 ↓ | C4 ↓ | Avg. ↑ |
|--------|---------|------|--------|
| FP16 | 6.65 | 10.86 | 60.64 |
| DuQuant | 10.78 | 14.33 | 51.90 |
| MoEQuant | 8.89 | 12.74 | 53.64 |
| EAQuant | 8.79 | 12.66 | 53.58 |
| ExpertQuant | **8.75** | **12.65** | **54.96** |

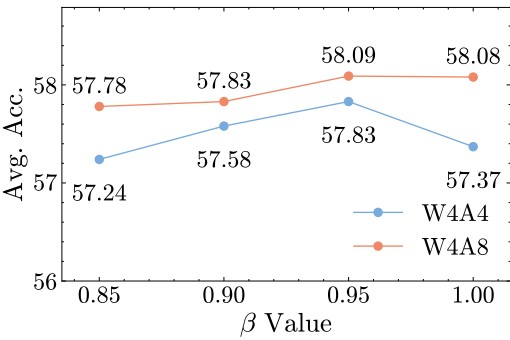

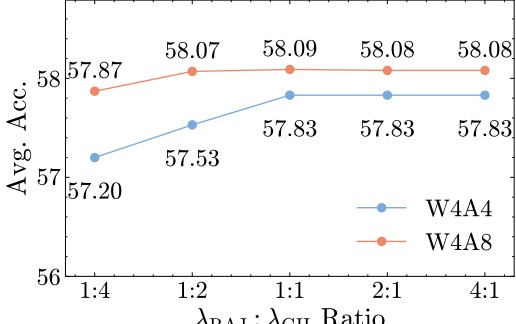

Figure 6: Average accuracy of different $\beta$ on OL-MoE under W4A4 and W4A8.

Figure 7: Effect of $\lambda_{\mathrm{RAJ}}$:$\lambda_{\mathrm{GH}}$ on OLMoE under W4A4 and W4A8.

Table 6: RAJ compared with KL divergence and cosine similarity on OLMoE (W4A4).

| Method | ARC-C | ARC-E | BoolQ | HellaS | OBQA | RTE | Wino. | Avg. ↑ |
|--------|-------|-------|-------|--------|------|-----|-------|--------|
| ES+GH+KL | 41.98 | 71.59 | **67.92** | 55.87 | 29.00 | 67.87 | **66.30** | 57.22 |
| ES+GH+Cosine | 42.66 | 71.17 | 67.13 | 55.77 | 30.60 | 66.79 | 65.43 | 57.08 |
| ES+GH+RAJ | **43.17** | **72.85** | 65.57 | **56.21** | **32.00** | **69.31** | 65.67 | **57.83** |

minor rank flips, whereas $\beta = 1.00$ spreads weight too uniformly and under-penalizes lower-rank mismatches. The peak at $\beta = 0.95$ indicates that a mild exponential decay strikes the right balance, prioritizing the top of the FP16 ranking while still regularizing lower ranks, thereby improving router consistency and downstream accuracy.

**Balancing Rank and Margin.** The router-loss is $\mathcal{L}_{\mathrm{router}}$ (Figure 7) is controlled by $\lambda_{\mathrm{RAJ}}$ and $\lambda_{\mathrm{GH}}$ to balance between rank and margin preservation. A sweep over ratios {1:4, 1:2, 1:1, 2:1, 4:1} on OLMoE shows that 1:1 attains the highest average accuracy under both W4A4 and W4A8. Skewing the weights toward *RAJ* (e.g., 4:1) over-penalizes benign rank shuffles and dampens margins, while favoring *GH* (e.g., 1:4) prioritizes separation but becomes less sensitive to near ties within the top-$k$. The symmetric setting aligns the objectives, improving match score and yielding the best downstream accuracy at the same bit budget.

**Perfect-Match Routing.** To isolate the effect of routing on downstream accuracy, the quantized model is executed under a *perfect-match* scheme in which the selected top-$k$ experts at every token exactly follow the FP16 router while all experts remain quantized. On OL-MoE at W4A4, this scheme boosts average accuracy across methods, +3.43% for DuQuant, +2.63% for MoEQuant, +1.71% for EAQuant, and +1.42% for ExpertQuant, validating that routing mismatches are a principal source of PTQ degradation (Figure 8). The smaller head-room for ExpertQuant is consistent with its higher match score, showing that *RAJ* and *GH* already recover much of the routing fidelity.

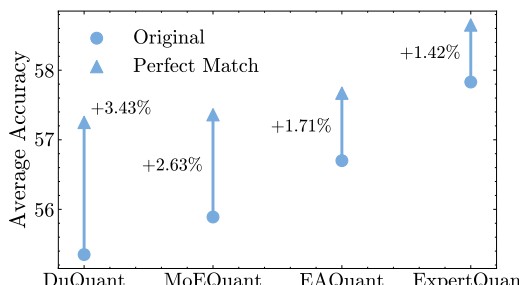

Figure 8: Average accuracy with "Perfect Match" routing, where the quantized model follows FP16 top-$k$ selections.

**RAJ Alignment.** To assess the role of our alignment choice, we replace RAJ with KL divergence and cosine similarity alignment, all computed strictly on the top-$k$ experts. Although these alternatives align logits within the selected set, they remain magnitude- or angle-based and are not inherently sensitive to ordering changes. RAJ, by contrast, directly compares the ordered top-$k$ sets with geometric rank weights and is invariant to affine logit transformations, making it responsive to

Table 7: Results on VLM benchmarks. The comparison includes MoEQuant, EAQuant, and ExpertQuant on Qwen3-VL-MoE (W4A4).

| Method | Accuracy ↑ | | | | | | | | |
|---|---|---|---|---|---|---|---|---|---|
| | GQA | ChartQA | ScienceQA | RWQA | K-DTC | OCR | MMMU | ai2d | Average |
| FP16 | 64.04 | 85.28 | 93.63 | 66.14 | 83.33 | 84.80 | 52.00 | 86.08 | 76.91 |
| DuQuant | 59.42 | 79.52 | 90.71 | 63.22 | 79.21 | 81.28 | 48.19 | 82.36 | 72.99 (-) |
| MoEQuant | 60.26 | 80.28 | 91.93 | 63.22 | 80.55 | 82.31 | 48.22 | **83.97** | 73.84 (1.17%) |
| EAQuant | 61.92 | 82.44 | 91.56 | 63.82 | 80.73 | 82.58 | 49.13 | 83.16 | 74.42 (1.96%) |
| ExpertQuant | **62.08** | **84.17** | **92.17** | **64.08** | **81.19** | **82.98** | **50.90** | 83.16 | **75.09 (2.88%)** |

the near-neighbor rank flips that dominate MoE routing errors. As a result, KL and cosine provide weaker guidance for stabilizing router rankings under quantization.

## 5 EXTENDING TO MULTIMODAL MoEs

To further demonstrate that our proposed quantization framework generalizes beyond language-only MoE models, we additionally evaluate it on a state-of-the-art multimodal architecture. Specifically, we conduct experiments on the **Qwen3-VL-30B-A3B-Instruct** model (Yang et al., 2025) to assess whether the same quantization strategies remain effective in vision–language settings.

For post-training calibration, we sample image–text pairs from the Flickr30k dataset (Plummer et al., 2015), following an analogous setup to our experiments. We quantize the model using the same configuration as in prior sections, with per-token activation quantization and per-channel weight quantization, and we evaluate W4A4 variant. Multimodal performance is assessed using `lmms-eval` (v0.5) under its default evaluation protocol. We test across a wide range of benchmarks covering visual reasoning, document understanding, scientific question answering, and real-world perception: GQA (Hudson & Manning, 2019), ChartQA (Masry et al., 2022), ScienceQA (Lu et al., 2022), RealWorldQA (Zhang et al., 2025), K-DTCBench (Ju et al., 2024), OCRBench (Liu et al., 2024c), MMMU (Yue et al., 2023), and AI2D Kembhavi et al. (2016).

Across all datasets, we observe that the W4A4 version of our method consistently outperforms competing quantization baselines. Notably, the gap is especially clear on multimodal reasoning and knowledge-intensive tasks. These findings validate that the proposed approach is not only effective for large MoE language models but also transfers robustly to more complex multimodal systems, highlighting its versatility and practical utility across diverse architectures.

## 6 CONCLUSION AND FUTURE WORK

This paper studies PTQ of MoE models and establishes that router performance governs low-bit accuracy, with errors concentrating as near-neighbor flips near the top-$k$ experts and arising from small score margins. We present ExpertQuant, a training-free, calibration-only framework that combines *ES* with *RAJ* and *GH* objectives to stabilize expert rankings and preserve margins; Empirically, the framework yields lower perplexity on C4 and WikiText-2 and higher zero-shot accuracy on OLMoE, DeepSeek-MoE, and Qwen3-MoE under W4A4 and W4A8. Looking ahead, we plan to extend our framework to multimodal MoE settings, where heterogeneous modalities may amplify router instability under quantization. We also aim to study adaptive precision guided by observed rank gaps and load patterns, as well as dynamic top-$k$ routing under quantization, to further enhance robustness and generalization across diverse tasks.

## REPRODUCIBILITY STATEMENT

We provide our implementation to ensure the reproducibility of our results in `https://anonymous.4open.science/r/expert_quant-code/`.

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

## A    MATCH SCORE FOR ROUTER CONSISTENCY

We quantify the agreement between FP16 and quantized routing with a *Match Score* that evaluates whether the quantized router preserves both the membership and the order of the FP16 top-$k$ experts. For each MoE layer $\ell \in \{1, \ldots, L\}$, let

$$I^{(\ell)} = (i_1^{(\ell)}, \ldots, i_k^{(\ell)}) = \text{top-}k(\mathbf{r}_\ell^{(\text{fp})}, k), \qquad J^{(\ell)} = (j_1^{(\ell)}, \ldots, j_k^{(\ell)}) = \text{top-}k(\mathbf{r}_\ell^{(\text{q})}, k),$$

denote the ordered top-$k$ expert indices from FP16 and quantized router logits, respectively, where $\text{top-}k(\cdot, k)$ returns the indices sorted by descending scores. Define the quantized rank function

$$\rho_\ell^{(\text{q})}(e) = \begin{cases} s, & \text{if } e = j_s^{(\ell)} \text{ for some } s \in \{1, \ldots, k\}, \\ \infty, & \text{if } e \notin J^{(\ell)}, \end{cases}$$

so that an FP16-selected expert missing from the quantized top-$k$ receives infinite rank (and thus zero credit). The layerwise agreement is then averaged over the $k$ FP16-selected experts and over all $L$ layers:

$$S = \frac{1}{kL} \sum_{\ell=1}^{L} \sum_{r=1}^{k} \frac{1}{1 + \big| r - \rho_\ell^{(\text{q})}(i_r^{(\ell)}) \big|}, \tag{10}$$

with the convention $1/(1 + \infty) = 0$. By construction, $S \in [0, 1]$; $S = 1$ if and only if the quantized top-$k$ exactly matches FP16 including ranks in every layer, and $S$ approaches 0 when FP16-selected experts are consistently absent from the quantized top-$k$ or appear only at much lower ranks. We compute $\mathbf{r}_\ell^{(\text{fp})}$ and $\mathbf{r}_\ell^{(\text{q})}$ for each input token and then report the dataset-level Match Score by averaging Equation (10) over tokens in a held-out set. Equivalently, if $S^{(t)}$ denotes Equation (10) evaluated on token $t$, the reported score is

$$\bar{S} = \frac{1}{T} \sum_{t=1}^{T} S^{(t)},$$

where $T$ is the number of evaluated tokens. This metric is sensitive to both rank inversions within the selected set and omissions of FP16-selected experts, which makes it a faithful proxy for router consistency under quantization.

## B    COMPREHENSIVE RESULTS

This appendix compiles the complete numerical results that support the figures and claims in the main text. Table 8 presents a module-wise ablation for OLMoE under W4A4, where leaving the router unquantized yields the largest average gain over fully W4A4, while without quantizing the attention projections ($\mathbf{q}/\mathbf{k}/\mathbf{v}$) has only marginal effects, indicating that routing is the most quantization-sensitive module in this setting. Table 9 reports results under W3A8 across OLMoE, DeepSeek-MoE, and Qwen3-MoE; on OLMoE, ExpertQuant attains the lowest perplexities and the highest average accuracy, improving over DuQuant by +5.89% and outperforming MoEQuant and EAQuant.

Sweeping the *RAJ* sharpness parameter $\beta$ in Table 10 shows performance peaking near $\beta{=}0.95$ for both W4A4 and W4A8, suggesting that moderately sharp rank penalties best stabilize router orderings without overfitting. Balancing the loss weights in $\mathcal{L}_{\text{router}} = \lambda_{\text{RAJ}}\mathcal{L}_{\text{RAJ}} + \lambda_{\text{GH}}\mathcal{L}_{\text{GH}}$ (Table 11) indicates that a 1:1 ratio between $\lambda_{\text{RAJ}}$ and $\lambda_{\text{GH}}$ consistently yields the best or near-best averages under both bit settings, highlighting the complementarity of rank and margin preservation. Finally, Table 12 evaluates an oracle "Perfect Match" that replaces each quantized router's top-$k$ expert indices with the FP16 selections at inference time while keeping all other modules unchanged; all methods benefit, reinforcing that preserving FP16 routing decisions is a principal driver of downstream accuracy and that our rank- and margin-aware objectives address the key failure mode.

## C    ADDITIONAL EXPERIMENTS

**Runtime Efficiency.**    We evaluate end-to-end decoding throughput (Tokens/Sec.) on a single NVIDIA A100 with batch size 1, using prompts of 1024 tokens and generating 128 tokens, and

Table 8: W4A4 on OLMoE; w/o leaves that module unquantized.

| Method | ARC-C | ARC-E | BoolQ | HellaS | OBQA | RTE | Wino. | Avg. |
|--------|-------|-------|-------|--------|------|-----|-------|------|
| W4A4 | 40.96 | 70.75 | 66.85 | 55.66 | 29.40 | 64.28 | 64.56 | 56.07 |
| w/o q | 39.24 | 70.45 | 66.54 | **56.05** | 29.40 | **66.06** | 65.27 | 56.14 (+0.14%) |
| w/o k | 39.76 | 70.26 | 66.38 | 55.84 | 29.40 | **66.06** | 65.19 | 56.13 (+0.11%) |
| w/o v | 40.96 | 70.75 | **66.85** | 55.57 | **30.80** | 64.32 | 64.56 | 56.26 (+0.34%) |
| w/o router | **42.49** | **71.17** | 66.42 | 55.97 | 30.60 | 65.70 | **65.98** | **56.90 (+1.50%)** |

Table 9: **Results under W3A8.** DuQuant serves as the baseline; the comparison includes MoE-Quant, EAQuant, and ExpertQuant on OLMoE, DeepSeek-MoE, and Qwen3-MoE.

| Model | Method | Perplexity ↓ | | Accuracy ↑ | | | | | | | |
|-------|--------|------|------|-------|-------|-------|--------|------|-------|-------|---------|
| | | Wiki2 | C4 | ARC-C | ARC-E | BoolQ | HellaS | OBQA | RTE | Wino. | Average |
| | FP16 | 6.65 | 10.86 | 46.59 | 77.10 | 70.09 | 58.47 | 32.60 | 71.12 | 68.51 | 60.64 |
| OLMoE | DuQuant | 10.78 | 14.33 | 36.00 | 66.71 | 61.74 | 50.76 | 27.60 | 61.37 | 59.12 | 51.90 (-) |
| | MoEQuant | 8.89 | 12.74 | 37.28 | 68.73 | 66.91 | 54.35 | 28.20 | 57.04 | 62.98 | 53.64 (+3.36%) |
| | EAQuant | 8.79 | 12.66 | 37.12 | 68.73 | 62.08 | 54.46 | 28.60 | 59.57 | 64.48 | 53.58 (+3.23%) |
| | ExpertQuant | **8.75** | **12.65** | **38.48** | **70.79** | **63.85** | **54.58** | **30.60** | **60.65** | **65.75** | **54.96 (+5.89%)** |

Table 10: $\beta$ in *RAJ* on OLMoE (W4A4 and W4A8).

| Bits | $\beta$-value | ARC-C | ARC-E | BoolQ | HellaS | OBQA | RTE | Wino. | Avg. ↑ |
|------|---------------|-------|-------|-------|--------|------|-----|-------|--------|
| **W4A4** | 0.85 | 42.22 | 72.22 | 65.38 | 56.08 | **32.20** | 66.72 | 65.88 | 57.24 |
| | 0.90 | 42.58 | 72.81 | **66.12** | **56.33** | 31.00 | 68.95 | 65.27 | 57.58 |
| | 0.95 | **43.17** | **72.85** | 65.57 | 56.21 | 32.00 | **69.31** | 65.67 | **57.83** |
| | 1.00 | 42.06 | 72.60 | 65.84 | 56.19 | 31.60 | 66.72 | **66.61** | 57.37 |
| **W4A8** | 0.85 | 41.81 | 72.98 | **67.71** | 57.13 | 31.00 | 66.06 | **67.80** | 57.78 |
| | 0.90 | **41.89** | **73.02** | **67.71** | 57.07 | 31.00 | 66.43 | 67.72 | 57.83 |
| | 0.95 | **41.89** | 72.85 | 67.68 | **57.16** | **31.40** | 68.59 | 67.09 | **58.09** |
| | 1.00 | **41.89** | 72.98 | **67.71** | **57.16** | 30.80 | **68.95** | 67.09 | 58.08 |

Table 11: $\lambda_{\text{RAJ}}$:$\lambda_{\text{GH}}$ on OLMoE (W4A4 and W4A8).

| Bits | $\lambda_{\text{RAJ}}$:$\lambda_{\text{GH}}$ | ARC-C | ARC-E | BoolQ | HellaS | OBQA | RTE | Wino. | Avg. ↑ |
|------|------|-------|-------|-------|--------|------|-----|-------|--------|
| **W4A4** | 1:4 | 42.41 | **73.57** | 65.05 | 55.95 | 31.20 | 67.15 | 65.04 | 57.20 |
| | 1:2 | 41.17 | 72.85 | **65.57** | **56.21** | **32.00** | **69.31** | 65.57 | 57.53 |
| | 1:1 | **43.17** | 72.85 | **65.57** | **56.21** | **32.00** | **69.31** | 65.67 | **57.83** |
| | 2:1 | **43.17** | 72.85 | **65.57** | **56.21** | **32.00** | **69.31** | 65.67 | **57.83** |
| | 4:1 | **43.17** | 72.85 | **65.57** | **56.21** | **32.00** | **69.31** | 65.67 | **57.83** |
| **W4A8** | 1:4 | 41.81 | **73.06** | 67.68 | 57.13 | 30.80 | 66.79 | **67.80** | 57.87 |
| | 1:2 | 41.55 | 72.98 | **67.89** | 57.13 | 31.20 | **68.95** | 66.77 | 58.07 |
| | 1:1 | **41.89** | 72.85 | 67.68 | **57.16** | **31.40** | 68.59 | 67.09 | **58.09** |
| | 2:1 | **41.89** | 72.98 | 67.71 | **57.16** | 30.80 | **68.95** | 67.09 | 58.08 |
| | 4:1 | **41.89** | 72.98 | 67.71 | **57.16** | 30.80 | **68.95** | 67.09 | 58.08 |

Table 12: "Perfect" replaces each quantized router's top-$k$ expert indices with the indices selected by the FP16 router at inference time, while keeping all other modules unchanged.

| Method | ARC-C | ARC-E | BoolQ | HellaS | OBQA | RTE | Wino. | Avg. |
|---|---|---|---|---|---|---|---|---|
| FP16 | 46.59 | 77.10 | 70.09 | 58.47 | 32.60 | 71.12 | 68.51 | 60.64 |
| DuQuant | 40.96 | 72.26 | 63.88 | 54.99 | 30.20 | 60.43 | 64.72 | 55.35 |
| DuQuant (perfect) | 42.66 | 74.75 | 66.88 | 55.69 | 31.80 | 64.98 | 64.01 | 57.25 |
| MoEQuant | 40.70 | 73.44 | 66.30 | 55.69 | 29.60 | 62.09 | 63.38 | 55.89 |
| MoEQuant (perfect) | 42.58 | 75.38 | 64.65 | 55.87 | 30.60 | 65.70 | 66.77 | 57.36 |
| EAQuant | 41.38 | 73.65 | 66.97 | 56.05 | 31.00 | 62.45 | 65.43 | 56.70 |
| EAQuant (perfect) | 43.09 | 74.54 | 66.91 | 55.93 | 31.60 | 64.98 | 66.61 | 57.67 |
| ExpertQuant | 43.17 | 72.85 | 65.57 | 56.21 | 32.00 | 69.31 | 65.67 | 57.83 |
| ExpertQuant (perfect) | 43.40 | 74.20 | 65.66 | 56.26 | 32.80 | 70.32 | 67.88 | 58.65 |

we report the mean $\pm$ std over five runs with the router fixed to W8A8. The Table 13 shows that DuQuant, MoEQuant, and EAQuant achieve similar runtimes with small standard deviations, indicating that ExpertQuant remains comparable even when incorporating heterogeneous expert scaling. This parity is expected because *ES* introduces only a lightweight per-expert scaling that is applied once during quantization and reduces to a constant multiplication at inference, incurring negligible overhead. Similarly, *RAJ* and *GH* are optimization objectives that are invoked only in the calibration stage to adjust quantization parameters, and thus do not alter the forward pass or add runtime cost. As a result, the inference computation graph and memory traffic remain identical to the baseline, ensuring that ExpertQuant preserves runtime efficiency while providing superior accuracy and perplexity.

Table 13: Tokens-per-second throughput under W4A8. Values are reported as mean$\pm$std over 5 runs.

| Model | Tokens-per-second $\uparrow$ | | |
|---|---|---|---|
| | OLMoE | DeepSeek | Qwen3 |
| DuQuant | $6.52 \pm 0.01$ | $23.88 \pm 0.05$ | $1.17 \pm 0.01$ |
| MoEQuant | $6.84 \pm 0.03$ | $24.56 \pm 0.06$ | $1.21 \pm 0.02$ |
| EAQuant | $6.72 \pm 0.05$ | $24.14 \pm 0.07$ | $1.20 \pm 0.01$ |
| ExpertQuant | $6.49 \pm 0.03$ | $23.79 \pm 0.06$ | $1.16 \pm 0.01$ |

**Weight-only quantization (W4A16).**    The main experiments in the paper primarily focus on scenarios where both weights and activations are quantized (W4A4, W4A8). To assess whether our method remains effective without activation quantization, we further conduct evaluations under the weight-only setting (W4A16). Compared against strong baselines such as AWQ (Lin et al., 2024b) and GPTQ (Frantar et al., 2022), our method consistently achieves large gains in zero-shot accuracy (See Table 14). This demonstrates that our approach not only addresses activation outliers but also provides robust improvements when only the weight domain is quantized.

**Varying calibration samples.**    Throughout the main experiments, the number of calibration samples is fixed at 256. To examine the sensitivity of our method to this choice, we additionally conduct experiments with 128 and 512 calibration samples (see Table 15 and 16). In both cases, our method continues to outperform all baselines by a clear margin. These results suggest that our method is robust across different calibration budgets and does not rely on a carefully chosen sample size to deliver improvements.

**Alternative calibration dataset.**    The primary results in the paper adopt C4 as the calibration dataset. To evaluate whether our method generalizes to other corpora, we also perform calibration on WikiText-2 (see Table 17). Even under this shift in data distribution, our method maintains

consistent advantages over all competing approaches. This highlights that our improvements are not tied to a specific calibration corpus and confirms the general applicability of our design.

Table 14: Zero-shot accuracy under W4A16 quantization.

| Model | Method | Accuracy ↑ | | | | | | | |
|-------|--------|------|------|-------|--------|------|------|------|---------|
| | | ARC-C | ARC-E | BoolQ | HellaS | OBQA | RTE | Wino. | Average |
| OLMoE | FP16 | 46.59 | 77.10 | 70.09 | 58.47 | 32.60 | 71.12 | 68.51 | 60.64 |
| | AWQ | 41.55 | 74.49 | 66.67 | 56.69 | 31.40 | **63.90** | 65.43 | 57.16 |
| | GPTQ | 42.66 | **75.21** | 67.34 | 57.33 | 31.80 | 62.82 | 67.72 | 57.84 |
| | ExpertQuant | **45.22** | **75.21** | **70.00** | **57.66** | **32.60** | 62.82 | **68.27** | **58.83** |
| DeepSeek | FP16 | 45.14 | 75.88 | 72.69 | 58.10 | 32.40 | 62.82 | 70.32 | 59.62 |
| | AWQ | 41.38 | 72.22 | 65.66 | 55.73 | 28.40 | 57.04 | 66.61 | 55.29 |
| | GPTQ | 42.49 | 73.61 | 67.55 | 56.21 | 29.40 | 60.43 | 65.67 | 56.48 |
| | ExpertQuant | **44.45** | **74.37** | **73.82** | **56.62** | **30.00** | **64.62** | **68.75** | **58.95** |
| Qwen3 | FP16 | 52.56 | 79.34 | 88.75 | 59.52 | 34.00 | 83.03 | 70.32 | 66.79 |
| | AWQ | 45.73 | 73.53 | 86.48 | 56.51 | **33.60** | 80.51 | 67.80 | 63.45 |
| | GPTQ | 48.81 | 76.94 | 87.95 | 57.01 | 32.80 | 77.98 | 68.82 | 64.33 |
| | ExpertQuant | **53.07** | **79.25** | **88.69** | **58.66** | 32.80 | **81.23** | **68.98** | **66.10** |

## D  ADDITIONAL VISUALIZATIONS

To complement the analyses in the main text, we provide extended visualizations for both router behavior and expert activations. First, confusion matrices for additional OLMoE layers are shown in Figures 9 and 10. These figures confirm our earlier observation that router errors after quantization are highly localized: most mis-selections occur between neighboring experts, with errors clustering near the diagonal rather than spreading arbitrarily. This further supports the claim that router performance is the dominant factor for MoE quantization.

Second, we present detailed activation distributions across experts at different depths. As shown in Figures 11 to 14, experts within the same layer exhibit markedly heterogeneous activation ranges. This reinforces the motivation behind our *Expert-Aware Scale*, which assigns each expert its own scaling factor instead of relying on max-aggregation across experts. These visualizations clearly illustrate why per-expert scaling avoids over-clipping light-tailed experts and prevents wasted quantization levels on the majority of channels.

Table 15: Zero-shot accuracy under W4A4 quantization with **128** calibration samples from **C4**.

| Model | Method | Accuracy ↑ | | | | | | | |
|-------|--------|------|------|-------|--------|------|------|------|---------|
| | | ARC-C | ARC-E | BoolQ | HellaS | OBQA | RTE | Wino. | Average |
| OLMoE | FP16 | 46.59 | 77.10 | 70.09 | 58.47 | 32.60 | 71.12 | 68.51 | 60.64 |
| | DuQuant | **40.53** | 70.24 | 65.20 | 55.20 | 29.80 | 62.82 | 63.06 | 55.26 |
| | MoEQuant | 39.85 | **71.59** | 65.81 | 54.86 | 29.60 | 62.82 | **65.43** | 55.71 |
| | EAQuant | 39.24 | 71.00 | **65.66** | 55.01 | 31.00 | 63.54 | 63.93 | 55.63 |
| | ExpertQuant | **39.76** | 71.00 | **65.66** | **55.81** | **31.80** | **66.79** | 64.17 | **56.43** |

Table 16: Zero-shot accuracy under W4A4 quantization with **512** calibration samples from **C4**.

| Model | Method | Accuracy ↑ | | | | | | | |
|-------|--------|------|------|-------|--------|------|------|------|---------|
| | | ARC-C | ARC-E | BoolQ | HellaS | OBQA | RTE | Wino. | Average |
| OLMoE | FP16 | 46.59 | 77.10 | 70.09 | 58.47 | 32.60 | 71.12 | 68.51 | 60.64 |
| | DuQuant | 38.74 | 69.74 | 65.66 | 55.20 | 29.80 | 63.54 | 63.85 | 55.22 |
| | MoEQuant | 40.53 | 72.10 | 63.67 | 54.96 | 28.40 | 65.70 | 64.56 | 55.70 |
| | EAQuant | 41.72 | 72.85 | **65.90** | **55.92** | 30.60 | 67.51 | **66.56** | 57.29 |
| | ExpertQuant | **42.66** | **74.62** | 65.63 | 55.91 | **31.60** | **69.31** | 66.46 | **58.03** |

Table 17: Zero-shot accuracy under W4A4 quantization with **256** calibration samples from **WikiText-2**.

| Model | Method | Accuracy ↑ | | | | | | | |
|-------|--------|-------|-------|-------|--------|------|------|------|---------|
| | | ARC-C | ARC-E | BoolQ | HellaS | OBQA | RTE | Wino. | Average |
| **OLMoE** | FP16 | 46.59 | 77.10 | 70.09 | 58.47 | 32.60 | 71.12 | 68.51 | 60.64 |
| | DuQuant | 39.16 | 69.82 | 64.37 | 55.01 | 27.40 | 62.45 | 62.19 | 54.34 |
| | MoEQuant | 39.60 | 72.18 | 67.80 | 55.30 | 28.40 | **63.18** | 65.75 | 56.03 |
| | EAQuant | 41.72 | 73.86 | 65.81 | 55.93 | 28.20 | 62.85 | 64.17 | 56.08 |
| | ExpertQuant | **42.49** | **73.95** | **68.01** | **56.02** | **28.60** | **63.18** | **65.90** | **56.88** |

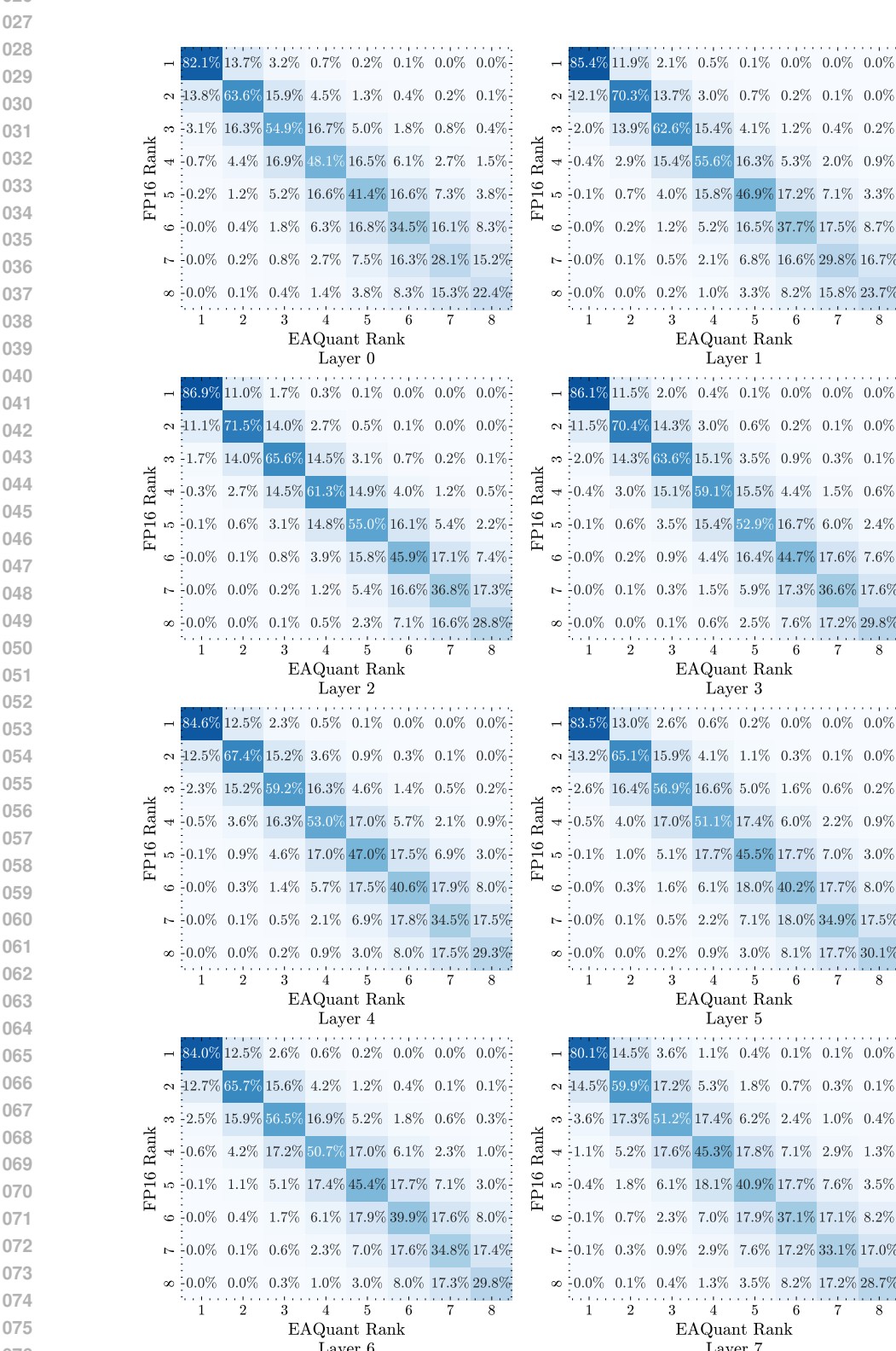

Figure 9: Confusion matrices at **layer 0-7** comparing FP16 vs. EAQuant top-$k$ indices under W4A4.

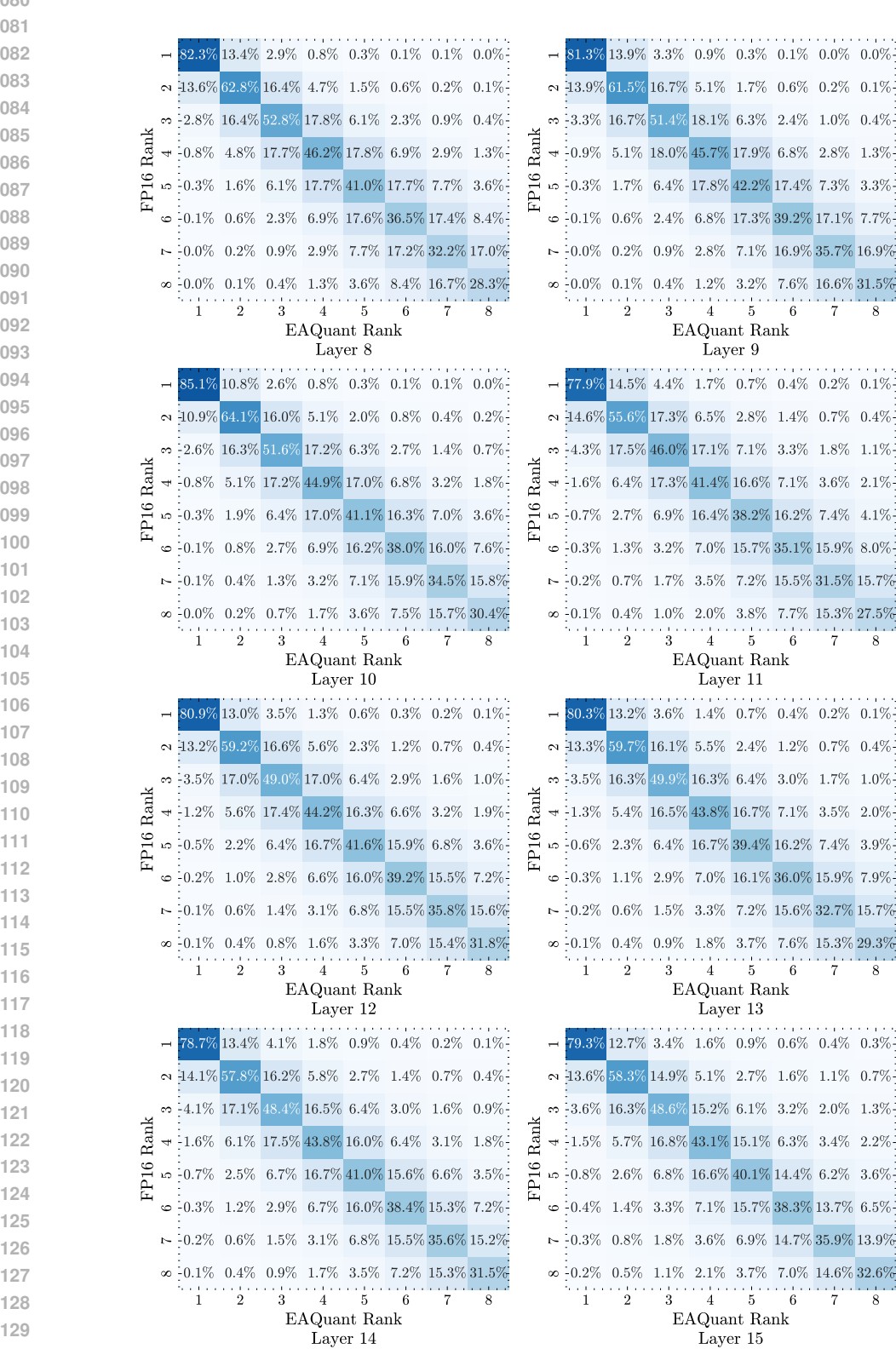

Figure 10: Confusion matrices at **layer 8-15** comparing FP16 vs. EAQuant top-$k$ indices under W4A4.

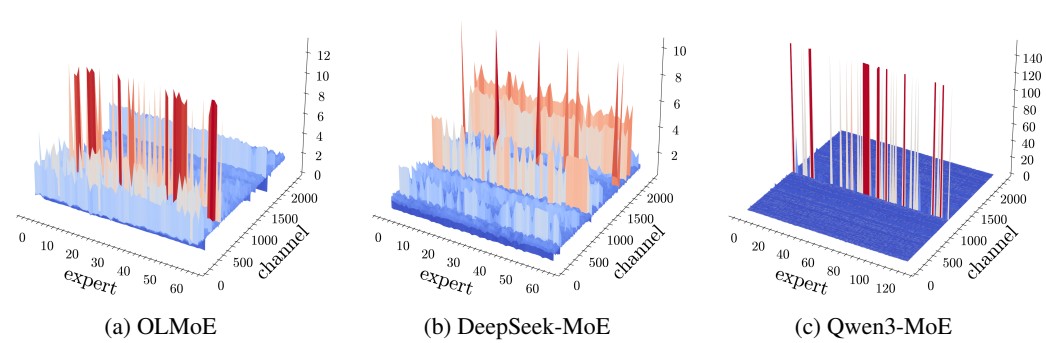

(a) OLMoE     (b) DeepSeek-MoE     (c) Qwen3-MoE

Figure 11: Expert activations in the **2nd layer** for each model.

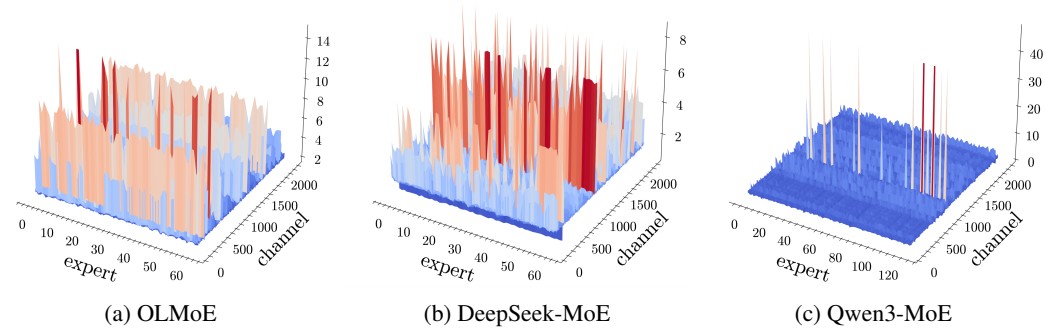

(a) OLMoE     (b) DeepSeek-MoE     (c) Qwen3-MoE

Figure 12: Expert activations in the **6th layer** for each model.

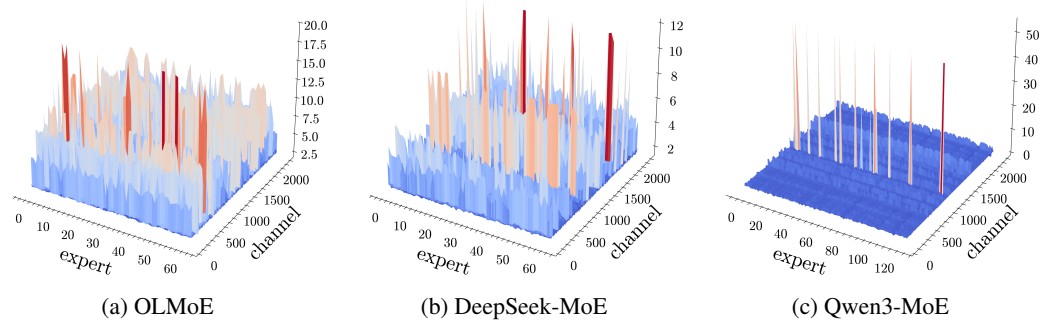

(a) OLMoE     (b) DeepSeek-MoE     (c) Qwen3-MoE

Figure 13: Expert activations in the **10th layer** for each model.

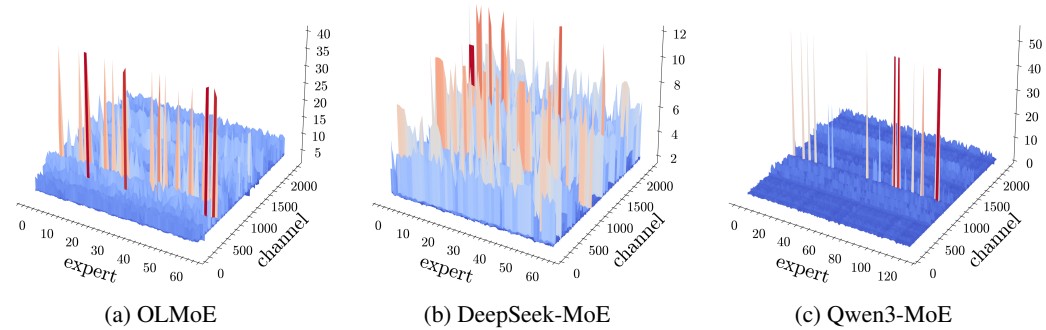

(a) OLMoE     (b) DeepSeek-MoE     (c) Qwen3-MoE

Figure 14: Expert activations in the **14th layer** for each model.

# E  THEORETICAL MOTIVATION FOR ROUTER STABILITY

In this section, we provide a theoretical explanation for why the *Rank-Aware Jaccard* (RAJ) and *Gap Hinge* (GH) losses are effective in solving the router stability issue.

**Recall.**  An MoE layer with $E$ experts and top-$k$ routing computes, for an input $x$:

$$y(x) \;=\; \sum_{j \in \text{top-}k} \pi_j(x)\, \mathcal{E}_j(x), \tag{11}$$

where $\pi_j(x)$ are (normalized) routing weights derived from $r(x)$, and $\mathcal{E}_j$ denotes the $j$-th expert. We consider a full-precision router $r^{(\text{fp})}$ and its quantized counterpart $r^{(q)}$. Let:

$$I = (i_1, \ldots, i_k) = \text{top-}k\big(r^{(\text{fp})}(x)\big), \tag{12}$$

$$J = (j_1, \ldots, j_k) = \text{top-}k\big(r^{(q)}(x)\big) \tag{13}$$

be the ordered top-$k$ expert indices for the full-precision and quantized routers, respectively. The corresponding MoE outputs are $y^{(\text{fp})}(x)$ and $y^{(q)}(x)$.

Quantizing router logits perturbs both the *set* and the *order* of the selected experts, as well as their routing weights. We show that the resulting MoE output error decomposes into: (i) a term that depends on how many top-$k$ *ranks* disagree between the full-precision and quantized routers, and (ii) a term that depends on weight discrepancies for experts whose ranks are preserved. This makes explicit why rank-aware alignment (RAJ) and margin enforcement (GH) directly targets the two components that govern quantization robustness.

## E.1  RANK-AWARE MOE ERROR DECOMPOSITION

We first define the number of rank mismatches:

$$d_{\text{rank}}(I, J) := \sum_{r=1}^{k} \mathbf{1}[\, i_r \neq j_r \,]. \tag{14}$$

**Lemma 1** (Rank-aware MoE error decomposition). *Assume* $\sup_j \|\mathcal{E}_j(x)\|_2 \leq C(x)$ *for some* $C(x) > 0$, *and let* $\pi_j^{(\text{fp})}$ *and* $\pi_j^{(q)}$ *denote the routing weights computed from* $r^{(\text{fp})}$ *and* $r^{(q)}$. *Then:*

$$\big\|y^{(q)}(x) - y^{(\text{fp})}(x)\big\|_2 \;\leq\; C(x) \left( 2\, d_{\text{rank}}(I, J) \;+\; \sum_{j \in I \cap J} \big|\pi_j^{(q)} - \pi_j^{(\text{fp})}\big| \right). \tag{15}$$

*Proof.*  We expand the difference:

$$y^{(q)}(x) - y^{(\text{fp})}(x) = \sum_{j \in J} \pi_j^{(q)} \mathcal{E}_j(x) \;-\; \sum_{j \in I} \pi_j^{(\text{fp})} \mathcal{E}_j(x) \tag{16}$$

$$= \sum_{j=1}^{E} \big(\pi_j^{(q)} \mathbf{1}[j \in J] - \pi_j^{(\text{fp})} \mathbf{1}[j \in I]\big) \mathcal{E}_j(x). \tag{17}$$

Apply the triangle inequality and the bound $\|\mathcal{E}_j(x)\|_2 \leq C(x)$:

$$\big\|y^{(q)}(x) - y^{(\text{fp})}(x)\big\|_2 \leq C(x) \sum_{j=1}^{E} \big|\pi_j^{(q)} \mathbf{1}[j \in J] - \pi_j^{(\text{fp})} \mathbf{1}[j \in I]\big|. \tag{18}$$

Define the disjoint sets:

$$S_1 := I \setminus J, \qquad S_2 := J \setminus I, \qquad S_3 := I \cap J.$$

Then the scalar sum in Equation (18) becomes:

$$\sum_{j=1}^{E} \left| \pi_j^{(q)} \mathbf{1}[j \in J] - \pi_j^{(\mathrm{fp})} \mathbf{1}[j \in I] \right| \tag{19}$$

$$= \sum_{j \in S_1} \pi_j^{(\mathrm{fp})} + \sum_{j \in S_2} \pi_j^{(q)} + \sum_{j \in S_3} \left| \pi_j^{(q)} - \pi_j^{(\mathrm{fp})} \right|. \tag{20}$$

Because routing weights satisfy $0 \leq \pi_j^{(\cdot)} \leq 1$, we have:

$$\sum_{j \in S_1} \pi_j^{(\mathrm{fp})} + \sum_{j \in S_2} \pi_j^{(q)} \leq |S_1| + |S_2| = |I \triangle J|, \tag{21}$$

where $I \triangle J$ is the symmetric difference. Since $I$ and $J$ are ordered lists of length $k$, each rank mismatch $i_r \neq j_r$ can contribute at most *two* elements to the symmetric difference (one from $I$, one from $J$). Hence:

$$|I \triangle J| \leq 2 \, d_{\mathrm{rank}}(I, J). \tag{22}$$

Substituting into Equation (18) gives:

$$\sum_{j=1}^{E} \left| \pi_j^{(q)} \mathbf{1}[j \in J] - \pi_j^{(\mathrm{fp})} \mathbf{1}[j \in I] \right| \leq 2 \, d_{\mathrm{rank}}(I, J) + \sum_{j \in I \cap J} \left| \pi_j^{(q)} - \pi_j^{(\mathrm{fp})} \right|.$$

Multiplying by $C(x)$ yields the desired bound. $\qquad\square$

Lemma 1 shows that router quantization error is governed by two factors:

1. **Rank disagreement:** number of top-$k$ positions differs between full-precision and quantization.
2. **Weight disagreement:** score perturbations for experts whose rank positions remain stable.

RAJ explicitly targets (1) by penalizing rank mismatches in a scale invariant way, while GH targets (2) by enforcing inter-rank *margins* that stabilizes the ordering under quantization noise.

## E.2 RAJ AS A RANK DISAGREEMENT

We now show that the Rank-Aware Jaccard (RAJ) loss directly targets the first term in Lemma 1, *i.e.*, the rank disagreement between the full-precision and quantized routers.

**Definition.** Recall that $(w_r)_{r=1}^{k}$ be a decreasing sequence of positive rank-weights (*e.g.*, $w_r = \beta^{r-1}$ with $0 < \beta \leq 1$) and define affinity vectors $A^{(\mathrm{fp})}, A^{(q)} \in \mathbb{R}^E$ by:

$$A^{(\mathrm{fp})}(e) := \begin{cases} w_r, & e = i_r, \\ 0, & \text{otherwise,} \end{cases} \qquad A^{(q)}(e) := \begin{cases} w_r, & e = j_r, \\ 0, & \text{otherwise.} \end{cases} \tag{23}$$

The rank-aware Jaccard similarity is:

$$J_W(I, J) := \frac{\sum_{e=1}^{E} \min \left( A^{(\mathrm{fp})}(e), A^{(q)}(e) \right)}{\sum_{e=1}^{E} \max \left( A^{(\mathrm{fp})}(e), A^{(q)}(e) \right)}, \qquad L_{\mathrm{RAJ}}(I, J) := 1 - J_W(I, J). \tag{24}$$

By construction, $J_W(I, J) = 1$ (hence $L_{\mathrm{RAJ}} = 0$) if and only if $I$ and $J$ are identical ordered lists.

We also define a rank-weighted disagreement similar to Equation (14):

$$d_{\mathrm{rank},w}(I, J) := \sum_{r=1}^{k} w_r \, \mathbf{1}[i_r \neq j_r], \tag{25}$$

which penalizes mismatches at higher ranks more strongly.

**Proposition 1** (RAJ controls rank-weighted disagreement). *Let $S_0 := \sum_{r=1}^{k} w_r$ and assume $w_r \in [w_{\min}, w_{\max}]$ with $0 < w_{\min} \leq w_{\max}$. Then there exist constants $c_1, c_2 > 0$ depending only on $w_r$ such that:*

$$c_1 \, d_{\text{rank},w}(I, J) \ \leq \ L_{\text{RAJ}}(I, J) \ \leq \ c_2 \, d_{\text{rank},w}(I, J). \tag{26}$$

*Proof.* Let:

$$S_{\min} := \sum_{e=1}^{E} \min\left(A^{(\text{fp})}(e), A^{(q)}(e)\right), \qquad S_{\max} := \sum_{e=1}^{E} \max\left(A^{(\text{fp})}(e), A^{(q)}(e)\right). \tag{27}$$

Each list contributes a total weight $S_0$; hence the union has total weight between $S_0$ (when $I = J$) and $2S_0$ (when they are disjoint), *i.e.*:

$$S_0 \ \leq \ S_{\max} \ \leq \ 2S_0. \tag{28}$$

We can therefore rewrite:

$$L_{\text{RAJ}} = 1 - \frac{S_{\min}}{S_{\max}} = \frac{S_{\max} - S_{\min}}{S_{\max}}, \tag{29}$$

which implies:

$$\frac{S_{\max} - S_{\min}}{2S_0} \ \leq \ L_{\text{RAJ}} \ \leq \ \frac{S_{\max} - S_{\min}}{S_0}. \tag{30}$$

We now relate $S_{\max} - S_{\min}$ to the rank-wise mismatches. Observe that at rank $r$ we either have $i_r = j_r$ (match) or $i_r \neq j_r$ (mismatch):

- If $i_r = j_r$, then both lists assign weight $w_r$ to the same expert at that position, so this contributes $w_r$ to both the intersection and the union at that location; it does not increase $S_{\max} - S_{\min}$.

- If $i_r \neq j_r$, then at least one of the lists assigns weight $w_r$ to an expert that the other does not assign weight $w_r$ to at that rank. In the worst case, these two experts are distinct and appear only once in each list; then this mismatch contributes at least $w_r$ to $S_{\max} - S_{\min}$ (the union counts both, while the intersection counts none). Conversely, because each position carries a weight at most $w_r$ on each side, the contribution of rank $r$ to $S_{\max} - S_{\min}$ is at most $2w_r$.

Thus there exist constants $a_1, a_2 > 0$ (in fact $a_1 = 1$, $a_2 = 2$ suffice) such that the contribution of rank $r$ to $S_{\max} - S_{\min}$ lies in $[a_1 w_r \mathbf{1}[i_r \neq j_r], a_2 w_r \mathbf{1}[i_r \neq j_r]]$. Summing over $r$ gives:

$$a_1 \, d_{\text{rank},w}(I, J) \ \leq \ S_{\max} - S_{\min} \ \leq \ a_2 \, d_{\text{rank},w}(I, J). \tag{31}$$

Combining Equation (30) and Equation (31), and using $S_{\max} \in [S_0, 2S_0]$, we obtain:

$$\frac{a_1}{2S_0} \, d_{\text{rank},w}(I, J) \ \leq \ L_{\text{RAJ}}(I, J) \ \leq \ \frac{a_2}{S_0} \, d_{\text{rank},w}(I, J). \tag{32}$$

Setting $c_1 = a_1/(2S_0)$ and $c_2 = a_2/S_0$ completes the proof. $\qquad\square$

**From rank-weighted mismatch to the Lemma 1 term.** Recall that the rank-unweighted mismatch is $d_{\text{rank}}(I, J) = \sum_{r=1}^{k} \mathbf{1}[i_r \neq j_r]$. Because $w_r \geq w_{\min}$ for all $r$, we have:

$$d_{\text{rank},w}(I, J) = \sum_{r=1}^{k} w_r \mathbf{1}[i_r \neq j_r] \ \geq \ w_{\min} \sum_{r=1}^{k} \mathbf{1}[i_r \neq j_r] = w_{\min} \, d_{\text{rank}}(I, J), \tag{33}$$

*i.e.*:

$$d_{\text{rank}}(I, J) \ \leq \ \frac{1}{w_{\min}} \, d_{\text{rank},w}(I, J) \ \leq \ \frac{1}{w_{\min} c_1} \, L_{\text{RAJ}}(I, J), \tag{34}$$

where we used the lower bound in Proposition 1.

Substituting this into Lemma 1, we obtain:

$$\left\| y^{(q)}(x) - y^{(\text{fp})}(x) \right\|_2 \ \leq \ C(x) \left( \frac{2}{w_{\min} c_1} L_{\text{RAJ}}(I, J) \ + \ \sum_{j \in I \cap J} \left| \pi_j^{(q)} - \pi_j^{(\text{fp})} \right| \right). \tag{35}$$

Thus, up to a constant factor depending only on the rank-weights $(w_r)$, the RAJ loss directly upper-bounds the *rank disagreement term* in the MoE error decomposition. Minimizing $L_{\text{RAJ}}$ therefore *provably reduces* the part of the quantization error that is due to misrouted experts and incorrect ordering of the top-$k$ list.

### E.3 GH AS A SURROGATE FOR WEIGHT DISAGREEMENT

We now connect the Gap Hinge (GH) loss to the second term in Lemma 1, *i.e.*, the routing weight disagreement $\sum_{j \in I \cap J} |\pi_j^{(q)} - \pi_j^{(\text{fp})}|$.

**Definition.** Let:

$$r_r^{(\text{fp})} := r_{i_r}^{(\text{fp})}, \qquad r_r^{(q)} := r_{i_r}^{(q)}, \qquad r = 1, \ldots, k, \tag{36}$$

be the logits of the top-$k$ experts under the two routers, restricted to the common index sequence $I$. Define the consecutive *margins*

$$\Delta_r^{(\text{fp})} := r_r^{(\text{fp})} - r_{r+1}^{(\text{fp})}, \qquad \Delta_r^{(q)} := r_r^{(q)} - r_{r+1}^{(q)}, \qquad r = 1, \ldots, k-1. \tag{37}$$

For the theoretical argument, it is convenient to consider the margin discrepancy:

$$d_{\text{gap}}(I) := \sum_{r=1}^{k-1} |\Delta_r^{(\text{fp})} - \Delta_r^{(\text{q})}|. \tag{38}$$

Our GH loss is defined as a scaled surrogate of $d_{\text{gap}}$:

$$L_{\text{GH}} = \frac{1}{k-1} \sum_{r=1}^{k-1} \left[ \Delta_r^{(\text{fp})} - \Delta_r^{(q)} - \gamma \right]_+, \tag{39}$$

with margin parameter $\gamma \geq 0$; when $\gamma = 0$ and the hinge is active near 0, $L_{\text{GH}}$ is proportional to the average margin discrepancy.

**Logit reconstruction from margins.** Up to an additive constant, the logit vectors $r^{(\text{fp})}$ and $r^{(q)}$ are fully determined by their consecutive margins. Since the softmax is invariant to adding a constant shift, we may, without loss of generality, recenter both vectors so that:

$$r_k^{(\text{fp})} = r_k^{(q)} = 0. \tag{40}$$

Then:

$$r_r^{(\text{fp})} = \sum_{s=r}^{k-1} \Delta_s^{(\text{fp})}, \qquad r_r^{(q)} = \sum_{s=r}^{k-1} \Delta_s^{(q)}, \qquad r = 1, \ldots, k. \tag{41}$$

Let $\varepsilon_s := \Delta_s^{(\text{fp})} - \Delta_s^{(q)}$ denote the margin errors. The logit error at rank $r$ is therefore:

$$e_r := r_r^{(\text{fp})} - r_r^{(q)} = \sum_{s=r}^{k-1} \varepsilon_s, \qquad r = 1, \ldots, k. \tag{42}$$

By the triangle inequality, we have:

$$|e_r| \leq \sum_{s=r}^{k-1} |\varepsilon_s|. \tag{43}$$

Let:

$$M := \max_{1 \leq r \leq k} \sum_{s=r}^{k-1} |\varepsilon_s|. \tag{44}$$

Then $|e_r| \leq M$ for all $r$, and hence:

$$\|e\|_2^2 = \sum_{r=1}^k e_r^2 \leq \sum_{r=1}^k M^2 = kM^2. \tag{45}$$

Taking square roots gives:

$$\|e\|_2 \leq \sqrt{k}\, M = \sqrt{k} \max_{1 \leq r \leq k} \sum_{s=r}^{k-1} |\varepsilon_s|. \tag{46}$$

Since for every $r$ the tail sum satisfies:

$$\sum_{s=r}^{k-1} |\varepsilon_s| \le \sum_{s=1}^{k-1} |\varepsilon_s|, \tag{47}$$

we obtain:

$$\|e\|_2 \le \sqrt{k} \sum_{s=1}^{k-1} |\varepsilon_s|. \tag{48}$$

By definition $d_{\mathrm{gap}}(I) = \sum_{s=1}^{k-1} |\varepsilon_s|$, so:

$$\left\| r^{(q)} - r^{(\mathrm{fp})} \right\|_2 \le \sqrt{k}\, d_{\mathrm{gap}}(I). \tag{49}$$

**Softmax Lipschitzness.** Assume that the routing weights over the top-$k$ experts are obtained by a softmax over the logits restricted to $I$:

$$\pi^{(\mathrm{fp})} = \mathrm{softmax}(r^{(\mathrm{fp})}), \qquad \pi^{(q)} = \mathrm{softmax}(r^{(q)}), \tag{50}$$

where both vectors are in $\mathbb{R}^k$ and we identify $\pi_r^{(\cdot)} := \pi_{i_r}^{(\cdot)}$.

The softmax mapping is smooth with a bounded Jacobian; in particular, there exists a constant $l_{\mathrm{sm}} > 0$ (depending only on $k$ and the logit range) such that:

$$\left\| \pi^{(q)} - \pi^{(\mathrm{fp})} \right\|_1 \le l_{\mathrm{sm}} \left\| r^{(q)} - r^{(\mathrm{fp})} \right\|_2. \tag{51}$$

Combining Equation (49) and Equation (51) yields:

$$\sum_{j \in I} \left| \pi_j^{(q)} - \pi_j^{(\mathrm{fp})} \right| = \left\| \pi^{(q)} - \pi^{(\mathrm{fp})} \right\|_1 \le l_{\mathrm{sm}} \sqrt{k}\, d_{\mathrm{gap}}(I). \tag{52}$$

**From $d_{\mathrm{gap}}$ to $L_{\mathrm{GH}}$.** If we choose the GH loss to be proportional to the average margin discrepancy, *i.e.*, $\gamma = 0$ and:

$$L_{\mathrm{G}} = \frac{1}{k-1} \sum_{r=1}^{k-1} |\Delta_r^{(q)} - \Delta_r^{(\mathrm{fp})}|, \tag{53}$$

then $d_{\mathrm{gap}}(I) = (k-1)L_{\mathrm{GH}}$, and Equation (52) becomes:

$$\sum_{j \in I} \left| \pi_j^{(q)} - \pi_j^{(\mathrm{fp})} \right| \le l_{\mathrm{sm}} \sqrt{k}(k-1)L_{\mathrm{G}}. \tag{54}$$

In particular, whenever $I = J$, the weight disagreement term in Lemma 1 is upper-bounded as:

$$\sum_{j \in I \cap J} \left| \pi_j^{(q)} - \pi_j^{(\mathrm{fp})} \right| = \sum_{j \in I} \left| \pi_j^{(q)} - \pi_j^{(\mathrm{fp})} \right| \le l_{\mathrm{sm}} \sqrt{k}(k-1)L_{\mathrm{G}}. \tag{55}$$

More generally, if we use the hinged version of Equation (39) with $\gamma > 0$, then as long as we operate in the regime were the hinge is active near 0 (so that $\left[ \Delta_r^{(q)} - \Delta_r^{(\mathrm{fp})} - \gamma \right]_+$ is comparable to $|\Delta_r^{(q)} - \Delta_r^{(\mathrm{fp})}|$), the same conclusion holds up to multiplicative constants.

**Connection to Lemma 1.** Combining Equation (54) with Lemma 1, and using RAJ to ensure $I \approx J$ (and in particular $I = J$ on most tokens), we obtain:

$$\left\| y^{(q)}(x) - y^{(\mathrm{fp})}(x) \right\|_2 \lesssim C(x) \left( 2\, d_{\mathrm{rank}}(I, J) + l_{\mathrm{sm}} \sqrt{k}(k-1)L_{\mathrm{GH}} \right), \tag{56}$$

where the first term is controlled by RAJ (Sec. E.2) and the second term is controlled by GH through the margin discrepancies. Intuitively, GH keeps the *relative gaps* between consecutive experts close to their full-precision counterparts; by softmax smoothness, this directly limits how much the routing weights over the shared top-$k$ experts can change, thereby addressing the second source of error in Lemma 1.

## F  USE OF LLMs

In preparing this paper, we used LLMs solely to aid in polishing the writing and improving clarity. The models were employed for grammar checking, smoothing sentence flow, and rephrasing for readability. All research ideas, methodological designs, experiments, and analyses were conceived and conducted entirely by the authors without reliance on LLMs.

