# OpenReview forum: "Router Choice Matters: Rank-Aware Post-Training Quantization for MoE Models"
_ICLR.cc/2026/Conference — ICLR 2026 Conference Withdrawn Submission_

### Official Review · Reviewer_QsxY · 2025-10-16

**Soundness:** 2
**Presentation:** 2
**Contribution:** 2
**Rating:** 4
**Confidence:** 4

**Summary:**

This paper focuses on the post-training quantization (PTQ) problem of Mixture-of-Experts (MoE) models. The authors point out that the main challenge of MoE quantization does not lie in the experts themselves, but in the error amplification effect of the router—once the router selects the wrong experts, the entire forward propagation is affected, leading to severe accuracy degradation. Therefore, preserving the router’s top-k ranking consistency and margin is crucial for low-bit MoE inference.
To address this, the authors propose ExpertQuant, a training-free, calibration-set-based PTQ framework that consists of three components: ES, RAJ, and GH. These modules work collaboratively to enhance routing consistency and downstream accuracy. Experiments on three MoE models show that ExpertQuant achieves significant performance improvements under both W4A8 and W4A4 quantization settings.

**Strengths:**

1. The work addresses the insufficiently explored issue of router ranking stability, filling an important gap in router optimization for MoE quantization.

2. The RAJ and GH losses function in a complementary manner, ensuring both correct expert selection and stable routing order.

3. The experimental evaluation is comprehensive, comparing multiple models across various datasets in terms of accuracy, and also reporting the match score to reflect the routing consistency with the full-precision model. The authors have provided reproducible code

**Weaknesses:**

1. The method appears to be a derivative of EAQuant, integrating two strategies that focus solely on router selection. However, the essence of MoE lies in the dynamic utilization of experts — intuitively, differences in expert usage frequency are likely to affect quantization performance. Despite this, the paper’s analysis remains limited to router quantization optimization, without addressing the variation in quantization effects across different experts, making the approach rather narrow.

2. The paper does not explain how the proposed methods influence the computation of quantization parameters. For instance, in MSE-based  approaches, it remains unclear how the introduced losses affect scale estimation. No concrete derivation or theoretical analysis is provided. From a code perspective, it seems that the authors have modified and optimized their work based on EAQuant. However, the authors have not conducted derivations regarding the feasibility of applying this approach to advanced methods such as AWQ or GPTQ.

3. The evaluation is restricted to language-based MoE models and does not extend to multimodal settings. Given that the heterogeneity of multimodal data may further exacerbate routing instability, it remains uncertain whether the proposed method can generalize and account for modality-specific influences on routing behavior.

**Questions:**

1. Could the paper provide a derivation showing how the proposed loss affects scale computation under standard min-max or MSE-based quantization, as well as theoretical integration with advanced methods such as GPTQ or AWQ? It would also be valuable to include ablation experiments comparing these variants.

2. The hyperparameters — α = 0.6 for ES, β = 0.95 for RAJ, and a loss weight ratio of 1:1 — are determined empirically through experimental search. Could the authors offer theoretical insights explaining why this combination performs best? Furthermore, if applied to different models, would these parameters need to be re-tuned, thereby increasing cross-model adaptation costs?

3. Could the paper clarify the boundary conditions of the RAJ method—for instance, whether RAJ remains effective when the number of experts is very large? Additionally, it would be helpful to justify the choice of RAJ similarity, and whether alternative measures such as cosine similarity or other metrics have been tested or considered.

---

> ### Author Response · Authors · 2025-11-18
> **Response to Reviewer QsxY**
>
> Dear Reviewer QsxY,
>
> Thank you for your thoughtful review and for highlighting our contributions to router ranking stability and routing-consistent PTQ. We appreciate your feedback regarding expert-level quantization behavior, theoretical analysis, and generalization to broader settings. We address each of your comments and questions in detail below.
>
> ---
>
> > **W1**: The method appears to be a derivative of EAQuant, integrating two strategies that focus solely on router selection. However, the essence of MoE lies in the dynamic utilization of experts — intuitively, differences in expert usage frequency are likely to affect quantization performance. Despite this, the paper’s analysis remains limited to router quantization optimization, without addressing the variation in quantization effects across different experts, making the approach rather narrow.
>
> We agree that experts can exhibit different activation frequencies and behaviors, and studying expert-specific quantization effects is indeed an interesting direction. However, this aspect is orthogonal to the core focus of our work.
>
> Our goal is to isolate and address the router-induced quantization errors, which we show, through both analysis and ablations, account for a significant portion of the degradation that remains even after expert FFNs are quantized. This perspective is fundamentally different from EAQuant, which focuses on expert-weight quantization and does not examine routing stability or rank-flip behavior.
>
> Because our contribution centers on the routing mechanism and its sensitivity under quantization, we intentionally focus on router-side errors rather than expert-level variance. The latter is complementary and falls outside the scope of this study, but we agree it is a valuable direction that can build on top of our findings.
>
> > **W2/Q1**: The paper does not explain how the proposed methods influence the computation of quantization parameters. For instance, in MSE-based approaches, it remains unclear how the introduced losses affect scale estimation. No concrete derivation or theoretical analysis is provided.
>
>
> We thank the reviewer for pointing this out. RAJ and GH are router losses used during (post-training) optimization that shape the quantized router logits.
>
> Concretely, in the revised manuscript we now add a rank-aware MoE error decomposition (Appendix E.1, Lemma 1) for the theoretical analysis:
>
> $$
> | y^{(q)}(x) - y^{(\mathrm{fp})}(x) |*2
> \le C(x)\Big(2,d_{\text{rank}}(I,J) * \sum_{j\in I\cap J}|\pi_j^{(q)} - \pi_j^{(\mathrm{fp})}|
>   \Big),
> $$
>
> where $d_{\text{rank}}(I,J)$ is the number of rank mismatches between the FP and quantized top-k expert lists, and $\pi_j^{(\cdot)}$ are the router weights. This shows that routing error is determined by:
>
> 1. Which experts (and ranks) are selected (first term), and
> 2. How the routing weights differ when the same experts are selected (second term).
>
> We then prove two properties:
>
> * **RAJ as the rank disagreement:**
> RAJ is a rank-aware Jaccard on the ordered top-k lists. In Appendix E.2, we show that RAJ is equivalent (up to constants) to a rank-weighted disagreement $d_{\text{rank},w}(I,J) = \sum_r w_r \mathbf{1}[i_r \neq j_r]$, and thus directly upper-bounds $d_{\text{rank}}(I,J)$. Importantly, RAJ is invariant to affine transforms of the logits, so it does not depend on the absolute scale; it only pushes the quantized router to preserve the FP16 ordering. This explains how the optimization driven by RAJ steers the quantized logits to the full-precision configurations.
> * **GH as the weight disagreement:**
> GH is defined on differences of differences (margins between consecutive experts). We show in Appendix E.3 that the consecutive gaps $\Delta_r = r_r - r_{r+1}$ uniquely determine the logits up to an additive constant, and, via softmax smoothness, margin discrepancies $|\Delta_r^{(fp)} - \Delta_r^{(\mathrm{r})}|$ yield a bound:
> $$
> \sum_{j\in I\cap J}|\pi_j^{(fp)} - \pi_j^{(\mathrm{r})}|
> \le \text{const} \cdot L_{\mathrm{GH}}.
> $$
> Thus GH explicitly controls the **second term** in the MoE error bound by keeping quantized margins close to the FP margins, which in turn stabilizes the routing weights for any MSE-optimal scale.
>
> Putting these together, the new analysis clarifies the role of RAJ and GH precisely by controlling the two terms in the bound above.

---

> > ### Author Response · Authors · 2025-11-18
> > **Response to Reviewer QsxY**
> >
> > > **W2/Q1**: From a code perspective, it seems that the authors have modified and optimized their work based on EAQuant. However, the authors have not conducted derivations regarding the feasibility of applying this approach to advanced methods such as AWQ or GPTQ.
> >
> >
> > Since our goal is to study the router in isolation rather than to combine or merge multiple quantization architectures, we build our implementation upon DuQuant, which provides a clean and widely adopted PTQ baseline (and is also the foundation used by EAQuant). This allows us to clearly isolate and evaluate the router-specific contributions of RAJ, GH, and ES.
> >
> > We also compare against AWQ and GPTQ in Appendix C (weight-only quantization) and show that our approach still achieves superior performance. Integrating our router-focused framework directly into full AWQ or GPTQ pipelines is certainly feasible, but it is orthogonal to the core problem we study and would require substantial additional engineering beyond the scope of this work. We consider this a promising direction for future work and will clarify this distinction in the revision.
> >
> > > **W3**: The evaluation is restricted to language-based MoE models and does not extend to multimodal settings. Given that the heterogeneity of multimodal data may further exacerbate routing instability, it remains uncertain whether the proposed method can generalize and account for modality-specific influences on routing behavior.
> >
> >
> > Thank you for the comment. To address this concern and demonstrate that our framework generalizes beyond language-only MoE models, we additionally evaluate it on Qwen3-VL-30B-A3B-Instruct. We follow an analogous calibration setup using image–text pairs from Flickr30k and apply the same W4A4 quantization configuration as in our experiments. Evaluation is conducted with lmms-eval across a broad suite of multimodal benchmarks including GQA, ChartQA, ScienceQA, RealWorldQA, K-DTCBench, OCRBench, MMMU, and AI2D.
> >
> >
> > | Method        | GQA   | ChartQA | ScienceQA | RWQA  | K-DTC | OCR   | MMMU  | ai2d  | Average |
> > |---------------|-------|---------|-----------|-------|-------|-------|-------|-------|---------|
> > | FP16      | 64.04 | 85.28 | 93.63 | 66.14 | 83.33 | 84.80 | 52.00 | 86.08 | 76.91 |
> > | DuQuant       | 59.42 | 79.52 | 90.71 | 63.22 | 79.21 | 81.28 | 48.19 | 82.36 | 72.99 (-) |
> > | MoEQuant      | 60.26 | 80.28 | 91.93 | 63.22 | 80.55 | 82.31 | 48.22 | **83.97** | 73.84 (1.17%) |
> > | EAQuant       | 61.92 | 82.44 | 91.56 | 63.82 | 80.73 | 82.58 | 49.13 | 83.16 | 74.42 (1.96%) |
> > | **ExpertQuant** | **62.08** | **84.17** | **92.17** | **64.08** | **81.19** | **82.98** | **50.90** | 83.16 | **75.09 (2.88%)** |
> >
> > Across all datasets, our method consistently outperforms existing baselines, with especially strong gains on multimodal reasoning and knowledge-intensive tasks. These results confirm that the proposed router-oriented quantization framework is not only effective for language MoE models but also transfers robustly to complex multimodal systems.
> >
> > We have added these experiments and the corresponding discussion as in Section 5.
> >
> > > **Q2**: The hyperparameters — α = 0.6 for ES, β = 0.95 for RAJ, and a loss weight ratio of 1:1 — are determined empirically through experimental search. Could the authors offer theoretical insights explaining why this combination performs best? Furthermore, if applied to different models, would these parameters need to be re-tuned, thereby increasing cross-model adaptation costs?
> >
> > We would like to clarify that our hyperparameter choices are general across models. As demonstrated in our experiments on multiple MoE architectures, the same settings (e.g., $\beta = 0.95$ and a $1:1$ loss ratio) work well without any model-specific retuning. For $\alpha$, we simply adopt the value used in prior work without additional search.
> >
> > Moreover, our ablation results show that a *range* of values still outperform baseline PTQ, indicating that the method is robust rather than sensitive. For interpretability, $\beta$ controls the strength of penalizing rank flips: very small values over-penalize and very large values under-penalize, while values near $0.95$ consistently preserve routing margins across models. Similarly, the $1:1$ loss ratio reflects the importance of balancing rank alignment and gap preservation rather than heavily weighting one objective.
> >
> > These observations suggest that our approach does not require precise or architecture-specific tuning, and the chosen hyperparameters are broadly applicable in practice.

---

> > > ### Author Response · Authors · 2025-11-18
> > > **Response to Reviewer QsxY**
> > >
> > > > **Q3-1**: Could the paper clarify the boundary conditions of the RAJ method—for instance, whether RAJ remains effective when the number of experts is very large?
> > >
> > > Thank you for the question. RAJ and GH operate **only on the top-k activated experts**, not on the full expert pool. Therefore, their effectiveness is determined by the stability of the *top-k margin*, and is fundamentally independent of whether the model has 64, 256, or 1,000+ total experts. Even in very large-MoE settings, only the top-k logits influence routing, so the number of non-selected experts does not affect our losses.
> > >
> > > While we are unable to run 1k-expert models due to resource constraints, our results already include SOTA models with **128 experts (e.g., Qwen3-MoE)** and show consistent improvements, supporting the scalability of our method.
> > >
> > > > **Q3-2**: Additionally, it would be helpful to justify the choice of RAJ similarity, and whether alternative measures such as cosine similarity or other metrics have been tested or considered.
> > >
> > > To evaluate the choice of RAJ similarity, we replaced RAJ with KL divergence and cosine similarity, computed strictly on the top-$k$ experts as shown in the below table.
> > >
> > > | Method        | ARC-C | ARC-E | BoolQ | HellaS | OBQA | RTE   | Wino. | Avg. ↑ |
> > > |---------------|-------|-------|-------|--------|------|-------|-------|--------|
> > > | ES+GH+KL      | 41.98 | 71.59 | **67.92** | 55.87 | 29.00 | 67.87 | **66.30** | 57.22 |
> > > | ES+GH+Cosine  | 42.66 | 71.17 | 67.13 | 55.77 | 30.60 | 66.79 | 65.43 | 57.08 |
> > > | ES+GH+RAJ     | **43.17** | **72.85** | 65.57 | **56.21** | **32.00** | **69.31** | 65.67 | **57.83** |
> > >
> > > While these alternatives align logits within the selected set, they are fundamentally magnitude- or angle-based and not inherently sensitive to ordering changes, which are the primary source of router errors under quantization. In contrast, RAJ explicitly compares the ordered top-$k$ sets with geometric rank weights and is invariant to affine logit transformations. This makes RAJ directly responsive to the near-neighbor rank flips that dominate MoE routing instability. As a result, KL and cosine provide weaker guidance for preserving routing consistency.
> > >
> > > We have added a new paragraph in Section 4.5 to discuss the choice of alignment.
> > >
> > > ---
> > >
> > > Thank you once again for your valuable feedback, which has significantly improved our work. We welcome further questions or suggestions for refinement.

---

> > ### Comment · Reviewer_QsxY · 2025-11-19
> > **Doubts About Routing Research**
> >
> > Grateful to authors for the timely supplements to both the theory and experiments. However, I still have the following doubts.
> >
> > > Our goal is to isolate and address the router-induced quantization errors, which we show, through both analysis and ablations, account for a significant portion of the degradation that remains even after expert FFNs are quantized.
> >
> > I don't think researching the quantization of MoE routing will have an essential impact on performance. Prior to this, I conducted relevant ablation experiments (on Qwen, DeepSeek, etc.) and found that when preserving full-precision expert selection, the routing choices are completely consistent with those in floating-point mode. However, the resulting PPL (Perplexity) still increases significantly. For example, in the Qwen-MoE architecture, the PPL on Wikitext2 rises from 8.07 to 8.36. If the router unit is also included in the quantization, the PPL only increases to 8.38. This 0.02 difference indicates that router quantization is not a key factor affecting MoE quantization. The focus of MoE quantization should be on the entire MoE block
> >
> > Additionally, I compared the outputs of full-precision and quantized MoE blocks and found that routing choices do not affect the output results. This might be because, due to routing quantization, the original ranking (e.g., ABCD) changes to a new one (e.g., HBJK). However, influenced by the inherent outputs and routing scores, the final output may not differ significantly from that of the full-precision version.
> >
> > Therefore, the paper only emphasizes the quantization of the router and does not conduct corresponding research on expert quantization. The author did not specify the method used for expert quantization (DuQuant?), if other methods are cited for expert quantization, the table should adopt a "plugin" approach to indicate the performance achieved by the combination. Claiming that its performance is "best" during the experimental phase somewhat overstates the importance of expert selection. Thus, I still believe that the innovation and contributions of this work are quite limited.
> >
> > >  RAJ and GH operate only on the top-k activated experts, not on the full expert pool. Therefore, their effectiveness is determined by the stability of the top-k margin, and is fundamentally independent of whether the model has 64, 256, or 1,000+ total experts. Even in very large-MoE settings, only the top-k logits influence routing, so the number of non-selected experts does not affect our losses.
> >
> > When E is extremely large, the proportion of top-k among all experts (k/E) approaches 0, and the probability that the quantization model selects the exact same top-k experts as the full-precision model decreases significantly. For example, if E increases from 100 to 10,000 while k is fixed at 4, the proportion of top-k drops from 4% to 0.04%, and the probability of "incorrectly selecting non-top-k experts" caused by quantization noise rises exponentially. RAJ relies on the Jaccard similarity (intersection/union) to measure consistency. When k/E is extremely small, even if the quantization model incorrectly selects only 1 expert, the size of the top-k intersection will drop sharply, the similarity value will approach 0, and the RAJ loss will tend to saturate (close to 1). Consequently, the gradient vanishes, making it impossible to effectively guide the optimization of quantization parameters. At this point, RAJ cannot distinguish between the difference of "incorrectly selecting 1 expert" and "incorrectly selecting k experts", and the optimization objective loses its guiding significance. Therefore, the design with β=0.95 is only suitable for medium-sized E, and the paper seems to lack consideration for this part.
> >
> > I’m not sure if my understanding is correct, or if I have any misunderstandings about the formula symbols in the paper.

---

> > > ### Author Response · Authors · 2025-11-19
> > > **Response to Reviewer QsxY**
> > >
> > > We thank the reviewer for the timely and thoughtful reply. Below, we address each of the raised points.
> > >
> > > ---
> > >
> > > > **Q1:**  I don't think researching the quantization of MoE routing will have an essential impact on performance. Prior to this, I conducted relevant ablation experiments... Thus, I still believe that the innovation and contributions of this work are quite limited.
> > >
> > > First, we would like to emphasize that our empirical findings are actually consistent with the reviewer’s observation that the change in perplexity is small. For example, in Table 2, the difference in PPL is indeed modest. However, similar perplexity does not imply identical or equivalent model behavior. This can be clearly seen from the accuracy benchmarks, which directly evaluate the **actual outputs** of the model rather than just the entropy-based score.
> > >
> > > Regarding the assertion that the outputs are effectively unchanged, we respectfully disagree. Even subtle differences in routing under quantization can lead to different token-level predictions on downstream tasks, which in turn affects benchmark scores and leads to the observed accuracy drop. Our results are precisely aimed at quantifying and analyzing this gap.
> > >
> > > As for expert quantization, we indeed adopt DuQuant **following the settings of prior work**. However, DuQuant itself does not introduce any MoE-specific design; it is a general-purpose quantization framework that simply treats expert FFNs as standard linear layers. Therefore:
> > >
> > > 1. Since our MoE quantization baselines are also based on DuQuant, our use of it ensures a fair and directly comparable setting.
> > > 2. Because DuQuant provides no special advantage tailored to MoE routing or expert structures, there is no hidden "extra" benefit from choosing it; this allows us to cleanly isolate and study the effect of routing quantization itself.
> > >
> > > Under this fair and widely adopted setup, our work identifies a concrete and previously underexplored issue: how routing quantization impacts MoE behavior and downstream accuracy, and proposes a targeted solution. We therefore believe our contributions are both novel and practically relevant, even though the effect may not be fully captured by perplexity alone.
> > >
> > > To make this clear, we have updated the experimental settings to explicitly state that our implementation builds on DuQuant, following the configurations used in prior work.
> > >
> > >
> > > > **Q2:** When E is extremely large, the proportion of top-k among all experts (k/E) approaches 0, ... Therefore, the design with β=0.95 is only suitable for medium-sized E, and the paper seems to lack consideration for this part.
> > >
> > > We would like to clarify that, both empirically and practically, we do not observe the saturation issue and "quantization noise rises exponentially" described by the reviewer. As shown in Figures 2, 9, and 10, the accuracy of the selected experts is typically very high (around 80%) for the first few experts and then gradually decreases to around 30%. This pattern is consistent across multiple architectures, including OLMoE with 64 experts and Qwen3 with 128 experts. In other words, the routing distribution remains highly concentrated on a relatively small subset of experts, and this behavior does not degrade as $E$ increases; in fact, it slightly improves in Figure 2 within the range we study. We would also like to emphasize that $\beta$ is applied only to the top-$k$ selected experts. Therefore, $\beta$ depends on $k$, not on the total number of experts $E$. This further ensures that the method remains stable even as $E$ grows.
> > >
> > > From a practical standpoint, for PTQ methods, the quantization procedure explicitly aims to preserve each layer’s activations by aligning the quantized outputs with the original full-precision activations. As a result, the router’s activation patterns are not drastically distorted, and the selected experts under quantization largely overlap with those in the original model. Pathological cases where the selected experts are almost completely non-overlapping, or where the routing becomes effectively saturated in an undesirable way, are therefore very unlikely, and we did not observe such behavior in our experiments.
> > >
> > >
> > > ---
> > >
> > > Thank you once again for your valuable feedback, which has significantly improved our work. We welcome further questions or suggestions for refinement.

---

> > > > ### Comment · Reviewer_QsxY · 2025-11-20
> > > >
> > > > Regarding the response to Q2, I have roughly understood the authors' perspective. From the standpoint of experiments and observations, extreme scenarios will not occur; however, I reserve my opinions on potential scenarios that may arise in the future.
> > > >
> > > > About Q1:
> > > > Firstly, enumerating the possible alignment of outputs under quantization is intended to demonstrate the importance of ranking hierarchy. This is because I found that the final output is derived from multiplying the ranking scores by the respective outputs. Taking the Qwen-MoE architecture as an example, the scores of the top 4 experts are 0.0684, 0.0590, 0.0537, and 0.0529 in sequence. Yet after weighted multiplication with their own outputs, the results become -0.0189, 0.0055, -0.0222, and 0.0317 respectively. In terms of output importance, the expert ranking shifts to 3, 4, 2, 1. This proves that after weighting, the proportion of each expert in the final output fails to maintain ranking consistency. Furthermore, if shared experts exist, their outputs will actually dominate the final result. Therefore, quantizing only the expert router seems insufficient for the quantization of the entire MoE architecture.
> > > >
> > > > Secondly, I would like to clarify that I only cited the perplexity (PPL) experiment as an example. In fact, I also conducted 2-bit, 4-bit, and 8-bit quantization solely on the router (even adopting the commonly used min-max method) on the HumanEval and GSM8K benchmarks. The results showed no significant performance degradation in experiments on Qwen and DeepSeek MoE models of different scales. On the contrary, quantizing the experts themselves leads to variations in their outputs, which in turn results in distinct final results and indeed exerts a substantial impact on precision. Additionally, in the paper, I noticed that the authors conducted ablation experiments in Figure 1, but these only involved q, k, v, and the router. Based on these experiments, they concluded the importance of keeping the router at full precision, yet overlooked that the FFN of the experts is the key to maintaining precision. The authors did not perform experiments related to the impact of ablation without experts (w/o expert), which is an oversight in their consideration.
> > > >
> > > > Meanwhile, due to the repetitive nature of the model structure, discrepancies in the output of the previous layer will lead to changes in the routing selection of the current layer. After all, the input to the router is still the hidden states output by the previous layer (which is essentially associated with the output of the entire decoder layer). Consequently, I hold the view that expert quantization is an unavoidable issue in MoE quantization. While researching router quantization is indeed a problem that needs to be addressed in MoE, it is insufficient when it comes to the quantization of the entire MoE architecture.

---

> > > > > ### Author Response · Authors · 2025-11-20
> > > > > **Response to Reviewer QsxY**
> > > > >
> > > > > We would like to again clarify that our approach **does not quantize only the router**. In all of our experiments, including those presented in Figure 1 and the main tables, **all experts are quantized**. This is explicitly stated in our settings, and the experimental pipeline in Figure 1 also reflects this design. Our findings regarding rank flipping among the top-k experts are therefore obtained *under full expert quantization*, and are fully consistent with the phenomenon observed by the reviewer.
> > > > >
> > > > > Regarding the performance degradation, we perform quantization on **all model components**, including the experts, and observe an accuracy drop of approximately 5% (as shown in Table 2). This degradation originates from the combined effect of quantizing both the router and the experts. In contrast, the reviewer’s example involves quantizing *only* the router, which naturally leads to minimal performance change.
> > > > >
> > > > > With respect to the ablations in Figure 1, we did not include a "w/o experts" setting because quantization is fundamentally aimed at reducing model size and accelerating inference. Expert weights constitute about 95% of the total model parameters in large MoE architectures. Under this context, an ablation that completely disables expert quantization would not match realistic PTQ deployment scenarios, and the resulting model would no longer achieve the intended compression or speedup. For this reason, we focus on settings where expert quantization is always enabled, which reflect real-world usage more faithfully.
> > > > >
> > > > > Finally, concerning the reviewer’s point about error propagation across layers: we agree that discrepancies in expert outputs can cause shifts in downstream routing decisions. This is precisely why expert quantization is included throughout our study. We emphasize again that **all** of our experiments are conducted on models where **experts are quantized**, and our conclusions about router sensitivity and ranking inconsistency are made *on top of* this full expert quantization. We therefore do not overlook the importance of expert FFNs; rather, our router-level findings highlight an *additional* quantization challenge that persists after experts have been quantized.
> > > > >
> > > > > In summary, expert quantization is fully incorporated in all of our experiments, and our investigation focuses on the residual routing-induced issues that remain after quantizing the entire MoE architecture.

---

> > > > > > ### Comment · Reviewer_QsxY · 2025-11-20
> > > > > >
> > > > > > I also want to emphasize again that I am not questioning whether the authors considered the quantization of the experts themselves in their experiments. Instead, **the paper only focuses on the quantization of the expert selection, making its overall contribution relatively limited.**
> > > > > >
> > > > > > > Contributions in paper
> > > > > >
> > > > > > > We demonstrate that the PTQ accuracy of MoE is primarily determined by router performance, which accounts for the majority of performance degradation.
> > > > > >
> > > > > > > We identify a router failure, i.e., near-neighbor rank flips, and propose Rank-aware Jaccard
> > > > > > Loss and Gap Hinge Loss to stabilize expert selection and preserve margins.
> > > > > >
> > > > > > > ...
> > > > > >
> > > > > > To verify the effectiveness, I have only performed partial validation due to time constraints. Quantizing the router network is merely an ablation comparison, **similar to what the authors presented in Figure 1** (observing the sensitivity of precision to the quantization of a specific component by leaving other components unquantized). This is not for deployment purposes but  **to identify issues**—by quantizing one component to examine its impact on the model after quantization. If the expert modules are quantized, it will result in a more severe drop in precision. Of course, I understand that the authors are not trying to argue with me about which component’s quantization is more important, and they have also mentioned: "...all of our experiments are conducted on models where experts are quantized..."
> > > > > >
> > > > > > Additionally, the authors mentioned that they quantized all model components (including the expert networks) and observed an accuracy drop of approximately 5%. However, within this 5% drop, have they clearly analyzed the impact of quantizing each individual component?
> > > > > >
> > > > > > I would like to **state again that I am not denying the necessity of router quantization**. At the same time, I acknowledge that the authors did quantize all weights in their experiments. Nevertheless, **in terms of the innovation and contribution of the overall method, it is relatively insufficient.**
> > > > > >
> > > > > > In the experimental phase, "expertquant" is used to represent the quantization effect of the entire network. **In fact, other quantization methods were also applied to the expert and self-attention modules, which are not mentioned in the paper**. Furthermore, most of the test sets in the experiments consist of multiple-choice questions, and it remains unclear how the method performs on math problems and coding tasks that require step-by-step reasoning.
> > > > > >
> > > > > > In summary, I currently intend to maintain my original score

---

> > > > > > > ### Author Response · Authors · 2025-11-25
> > > > > > > **Response to Reviewer QsxY**
> > > > > > >
> > > > > > > Thank you again for the detailed and thoughtful response.
> > > > > > >
> > > > > > > ---
> > > > > > >
> > > > > > > Regarding the reported 5% performance reduction, our method achieves an improvement of approximately 2-3% by specifically addressing router quantization. This demonstrates that managing router-related errors is an effective and complementary way to enhance overall performance, without conflicting with other components of the model. While fully disentangling the sources of the 5% drop is challenging, given that the effects of quantizing different parts of the model are interdependent, we do provide a related component-level analysis in Figure 1 to help illuminate these interactions.
> > > > > > >
> > > > > > > Additionally, We have updated our paper to clearly state that our implementation is based on DuQuant. We believe this clarification helps distinguish our focus on router-specific issues and provides better context for readers.
> > > > > > >
> > > > > > > Finally, for the reviewer's concerns regarding potential "similar outputs" for math and coding, we conducted further experiments using Qwen3-30B-A3B under W4A4 settings. The comparison between FP16, DuQuant, and our methods shows that router management yields consistent improvements, and importantly, we do not observe the issues mentioned by the reviewer.
> > > > > > >
> > > > > > > | Task        | GSM8K (8-shot CoT EM) | HumanEval (0-shot Pass@1) |
> > > > > > > |-------------|-------------------------|-----------------------------|
> > > > > > > | FP16        | 87.89                  | 41.70                        |
> > > > > > > | DuQuant     | 75.72                  | 32.34                       |
> > > > > > > | MoEQuant    | 77.12                  | 35.83                       |
> > > > > > > | EAQuant     | 78.25                  | 37.20                        |
> > > > > > > | ExpertQuant | 80.21                  | 38.41                       |
> > > > > > >
> > > > > > > ---
> > > > > > >
> > > > > > > We sincerely appreciate the reviewer’s time, effort, and constructive feedback. We fully respect the scoring criteria and hope that our clarifications and additional results help address the reviewer’s concerns and our paper's positioning.

---

### Official Review · Reviewer_tXdu · 2025-10-28

**Soundness:** 3
**Presentation:** 3
**Contribution:** 2
**Rating:** 4
**Confidence:** 4

**Summary:**

This paper studies PTQ for MoE and identifies the router (expert score/rank mismatch with full-precision model) as the main source of quantization-induced accuracy loss. The authors propose ExpertQuant, a PTQ framework optimized for expert rank maintaining. Experiments on OLMoE, DeepSeek-MoE, and Qwen3-MoE show consistent improvements in perplexity and zero-shot accuracy under low-bit settings.

**Strengths:**

1. The paper's core strength lies in its empirical validation that router instability is the main bottleneck in MoE quantization. The specific insight that errors are dominated by "near-neighbor rank flips" is a new perspective.
2. The framework's effectiveness is demonstrated across multiple state-of-the-art MoE models and a wide range of benchmarks. The paper includes thorough ablation studies and a "Perfect Match" routing experiment, which strongly support the central hypothesis.

**Weaknesses:**

1. Requires careful tuning of hyperparameters (e.g. $\beta, \lambda$ ratios), which limits generalization and usability.
2. Lacks theoretical explanation of why rank preservation leads to accuracy recovery.
3. Lacks of efficiency experiments.
4. Most of the ideas in this work have appeared in EAQuant, which weakens the contribution of this work to some extent.

**Questions:**

1. Why use channel smoothing instead of rotation-based approaches (e.g. Hadamard transform)  to alleviate outlier problems? I think roation-based methods don't have the expert-conflict problem, which was seen as a challenge in this work("EXPERT-AWARE SCALE" is proposed to solve this).
2. The router is typically not quantized in MoE quantization. Figure 1 is confusing.

---

> ### Author Response · Authors · 2025-11-18
> **Response to Reviewer tXdu**
>
> Dear Reviewer tXdu,
>
> Thank you for your constructive review and for acknowledging our empirical findings on router instability and the comprehensive evaluation across multiple MoE models. Your comments on hyperparameter tuning, theoretical grounding, efficiency considerations, and the relation to EAQuant are highly appreciated. We address each of these points and your specific questions in the responses below.
>
> ---
>
> > **W1**: Requires careful tuning of hyperparameters (e.g. $\beta$, $\lambda$ ratios), which limits generalization and usability.
>
> We would like to clarify that our method does **not** require careful hyperparameter tuning in practice. As shown in our experiments across multiple MoE architectures, the same settings (e.g., $\beta = 0.95$ and a $1:1$ loss ratio) generalize well without retuning.
>
> Moreover, our ablation figures demonstrate that a *range* of values still outperforms baseline PTQ, indicating that the method is **robust** rather than sensitive. For interpretability, $\beta$ controls how aggressively we penalize rank flips: very small $\beta$ over-penalizes and very large $\beta$ under-penalizes, but values near $0.95$ consistently work across models. Similarly, the $1:1$ loss ratio reflects the importance of balancing rank alignment and gap preservation rather than heavily weighting one objective.
>
> These observations suggest that our approach does **not** rely on precise or model-specific tuning, and the chosen hyperparameters are broadly applicable.
>
>
> > **W2**: Lacks theoretical explanation of why rank preservation leads to accuracy recovery.
>
> We thank the reviewer for pointing this out. RAJ and GH are router losses used during (post-training) optimization that shape the quantized router logits.
>
> Concretely, in the revised manuscript we now add a rank-aware MoE error decomposition (Appendix E.1, Lemma 1) for the theoretical analysis:
>
> $$
> | y^{(q)}(x) - y^{(\mathrm{fp})}(x) |*2
> \le C(x)\Big(2,d_{\text{rank}}(I,J) * \sum_{j\in I\cap J}|\pi_j^{(q)} - \pi_j^{(\mathrm{fp})}|
>   \Big),
> $$
>
> where $d_{\text{rank}}(I,J)$ is the number of rank mismatches between the FP and quantized top-k expert lists, and $\pi_j^{(\cdot)}$ are the router weights. This shows that routing error is determined by:
>
> 1. Which experts (and ranks) are selected (first term), and
> 2. How the routing weights differ when the same experts are selected (second term).
>
> We then prove two properties:
>
> * **RAJ as the rank disagreement:**
> RAJ is a rank-aware Jaccard on the ordered top-k lists. In Appendix E.2, we show that RAJ is equivalent (up to constants) to a rank-weighted disagreement $d_{\text{rank},w}(I,J) = \sum_r w_r \mathbf{1}[i_r \neq j_r]$, and thus directly upper-bounds $d_{\text{rank}}(I,J)$. Importantly, RAJ is invariant to affine transforms of the logits, so it does not depend on the absolute scale; it only pushes the quantized router to preserve the FP16 ordering. This explains how the optimization driven by RAJ steers the quantized logits to the full-precision configurations.
> * **GH as the weight disagreement:**
> GH is defined on differences of differences (margins between consecutive experts). We show in Appendix E.3 that the consecutive gaps $\Delta_r = r_r - r_{r+1}$ uniquely determine the logits up to an additive constant, and, via softmax smoothness, margin discrepancies $|\Delta_r^{(fp)} - \Delta_r^{(\mathrm{r})}|$ yield a bound:
> $$
> \sum_{j\in I\cap J}|\pi_j^{(fp)} - \pi_j^{(\mathrm{r})}|
> \le \text{const} \cdot L_{\mathrm{GH}}.
> $$
> Thus GH explicitly controls the **second term** in the MoE error bound by keeping quantized margins close to the FP margins, which in turn stabilizes the routing weights for any MSE-optimal scale.
>
> Putting these together, the new analysis clarifies the role of RAJ and GH precisely by controlling the two terms in the bound above.
>
> > **W3**: Lacks of efficiency experiments.
>
> We would like to clarify that the runtime behavior of ES has been evaluated in Appendix C: *Runtime Efficiency*.
>
> As shown in Appendix C, ES introduces only negligible runtime overhead. The token/s measurements below demonstrate that ES does not incur extra overhead during inference and therefore requires no trade-off or additional tuning, even when the number of experts grows.
>
> | Model       | OLMoE       | DeepSeek     | Qwen3       |
> | ----------- | ----------- | ------------ | ----------- |
> | DuQuant     | 6.52 ± 0.01 | 23.88 ± 0.05 | 1.17 ± 0.01 |
> | MoEQuant    | 6.84 ± 0.03 | 24.56 ± 0.06 | 1.21 ± 0.02 |
> | EAQuant     | 6.72 ± 0.05 | 24.14 ± 0.07 | 1.20 ± 0.01 |
> | ExpertQuant | 6.49 ± 0.03 | 23.79 ± 0.06 | 1.16 ± 0.01 |

---

> > ### Author Response · Authors · 2025-11-18
> > **Response to Reviewer tXdu**
> >
> > > **W4**: Most of the ideas in this work have appeared in EAQuant, which weakens the contribution of this work to some extent.
> >
> >
> > We respectfully clarify that our work is substantially different from EAQuant in both motivation and technical contribution.
> >
> > Our central contribution is identifying **router-induced degradation**, a factor that prior work does not address. A key finding of this paper is that the router, despite accounting for less than 0.03% of parameters, can contribute disproportionately to MoE quantization error through **top-$k$ rank instability**. We analyze, quantify, and mitigate this phenomenon **for the first time**. Building on this insight, we propose **RAJ** and **GH**, which directly optimize router logits to preserve the FP16 top-$k$ set and margin structure under quantization. No analogous objectives or ideas exist in prior MoE quantization work. By contrast, EAQuant focuses exclusively on expert FFN quantization and does not examine routing errors, rank flips, or expert-selection stability.
> >
> > In summary, although both works address MoE quantization, the research questions, targeted components, and methodological innovations are entirely different. Our work introduces the first **router-focused quantization framework** and identifies a previously overlooked source of degradation that EAQuant does not cover.
> >
> >
> > > **Q1**: Why use channel smoothing instead of rotation-based approaches (e.g. Hadamard transform) to alleviate outlier problems? I think roation-based methods don't have the expert-conflict problem, which was seen as a challenge in this work("EXPERT-AWARE SCALE" is proposed to solve this).
> >
> >
> > Thank you for the question. We would like to clarify that Expert-Aware Scale (ES) is not the main focus of our work, but rather a secondary observation that helps stabilize expert activations. Our primary contribution and the majority of our analysis center on router-induced errors and how to mitigate them through the proposed RAJ and GH losses. Consequently, we did not pursue an extensive exploration of alternative outlier-mitigation methods in this paper.
> >
> > That said, rotation-based approaches are indeed effective for handling activation outliers in dense models, and we agree they could also be applied to MoE experts. Exploring rotations or hybrid strategies for expert outlier control is a promising direction, and we will consider it as future work. Our current ES method is intended as a simple and practical solution that supports our router-focused framework, rather than a complete replacement for all rotation-based techniques.
> >
> >
> > > **Q2**: The router is typically not quantized in MoE quantization. Figure 1 is confusing.
> >
> >
> > We would like to clarify that quantizing the router is a common practice in existing MoE quantization frameworks. Prior works such as EAQuant quantize the router by default, and widely used open-source PTQ toolkits do the same. For example, both GPTQModel and AWQ explicitly quantize the router module in their MoE implementations. We provide the link below. Given this, quantizing the router is *not* unusual or confusing; it reflects how MoE PTQ is done in practice. Our Figure 1 follows this standard setup, allowing us to isolate and study the router’s impact on quantization accuracy.
> >
> > * GPTQModel (Qwen3-MoE): [link to code](https://github.com/ModelCloud/GPTQModel/blob/40759cdf06c17ea6c57637fa075f8c10547b4a6f/gptqmodel/models/definitions/qwen3_moe.py#L35C9-L35C22)
> > * AWQ (Qwen3-MoE) [link to code](https://github.com/casper-hansen/AutoAWQ/blob/88e4c76b20755db275574e6a03c83c84ba3bece5/awq/models/qwen3_moe.py#L53)
> >
> > ---
> >
> > Thank you once again for your valuable feedback, which has significantly improved our work. We welcome further questions or suggestions for refinement.

---

> ### Comment · Reviewer_tXdu · 2025-11-20
>
> Thanks for the author's reply.
>
> 1. If the router is so sensitive, why not leave it unquantized? I think it's more straightforward.
> 2. The sensitivity of router has been widely discussed in the community and has almost become a consensus[3]. The two links provided by the author don't prove anything; we can easily find many compressor that do not quantize MoE router as well.[1] [2]
>
> Overall, the motivation of this paper is weird to me.
>
> [1] https://github.com/vllm-project/llm-compressor/blob/a270f33a17705dfd9e1520aae534ffff83ddfe15/examples/model_free_ptq/kimi_k2_thinking_fp8_block.py#L14
>
> [2] https://docs.vllm.ai/projects/llm-compressor/en/latest/examples/quantizing_moe/#step-1-select-a-model-dataset-and-recipe
>
> [3] Ma, Wenhan, et al. "Stabilizing MoE Reinforcement Learning by Aligning Training and Inference Routers." arXiv preprint arXiv:2510.11370 (2025).

---

> > ### Author Response · Authors · 2025-11-20
> > **Response to Reviewer tXdu**
> >
> > We thank the reviewer for the additional comments and for sharing the related references.
> >
> > ---
> > We agree that keeping the router in full precision is a valid design choice, and indeed some existing systems adopt this approach. However, our work specifically addresses the complementary setting: when the router is quantized, how can we mitigate the resulting accuracy degradation? In fact, prior MoE quantization studies that we use as baselines also operate under this setting. Therefore, our method is not in conflict with approaches that keep the router in FP; rather, it provides a solution for scenarios where router quantization is required.
> >
> > Since both design choices are used in practice, we believe it is valuable to study router quantization instead of assuming it will always remain unquantized. Our contribution should be understood as addressing the scenario where:
> >
> > - the entire model are compressed, and
> > - existing methods suffer from output degradation due to routing instability.
> >
> > Finally, we would like to emphasize that research progresses by exploring diverse possibilities. Both FP-router and quantized-router settings are meaningful in different deployment contexts, and we believe that examining the quantized-router scenario is worthwhile and contributes constructively to the broader MoE quantization landscape.
> >
> > ---
> >
> > Thank you once again for your valuable feedback, which has significantly improved our work. We welcome further questions or suggestions for refinement.

---

### Official Review · Reviewer_XAzX · 2025-10-31

**Soundness:** 2
**Presentation:** 3
**Contribution:** 2
**Rating:** 4
**Confidence:** 4

**Summary:**

This study centers on the post-training quantization (PTQ) challenge specific to Mixture-of-Experts (MoE) models. The researchers highlight that the primary difficulty in quantizing MoE models is not the experts themselves, but rather the router’s tendency to amplify errors. If the router makes incorrect expert selections, it disrupts the entire forward propagation process, resulting in substantial drops in model accuracy. Thus, maintaining the router’s top-k ranking consistency and the margin between rankings is essential for enabling low-bit inference of MoE models. To tackle this issue, the authors introduce ExpertQuant—a PTQ framework that requires no additional training and relies on a calibration dataset. This framework comprises three key components: ES, RAJ, and GH. These modules operate in tandem to improve both the consistency of routing decisions and the model’s downstream task accuracy. Testing across three distinct MoE models demonstrates that ExpertQuant delivers notable performance gains when operating under both W4A8 and W4A4 quantization configurations.

**Strengths:**

1. The paper identifies a critical yet underexplored bottleneck in MoE model quantization: router-induced error cascading rather than expert-level inefficiencies. Through controlled studies (e.g., Figure 1, which shows preserving the router in full precision yields ~1.50% higher accuracy than unquantizing other modules like attention projections) and confusion matrix analyses (Figures 2, 9, 10), it empirically demonstrates that quantization errors primarily manifest as "near-neighbor rank flips" around top-k experts.
2. The RAJ and GH losses operate in a complementary fashion, working together to guarantee both the accuracy of expert selection and the stability of routing order for MoE models. The paper’s experimental validation is thorough: it assesses performance across multiple MoE architectures, evaluates accuracy on diverse datasets, and incorporates the Match Score as a metric to quantify how well the quantized router’s decisions align with those of the full-precision model. Additionally, the authors have made reproducible code available to support the transparency and replicability of their findings.

**Weaknesses:**

1. The paper focuses heavily on optimizing router quantization (via RAJ and GH losses) but overlooks a core attribute of MoE architectures: variations in expert usage frequency and their impact on quantization performance. As the critique notes, MoE’s essence lies in dynamic expert utilization, yet the study provides no analysis of how differing expert usage patterns (e.g., frequently vs. rarely activated experts) affect quantization efficacy. It only addresses router-related errors and does not explore or mitigate disparities in quantization effects across individual experts—leaving a gap in its coverage of MoE-specific challenges and narrowing the method’s comprehensiveness.
2. The paper’s evaluation is limited to language-only MoE models (OLMoE, DeepSeek-MoE, Qwen3-MoE) and does not extend to multimodal MoE settings. Given that multimodal data heterogeneity could exacerbate routing instability, the method’s ability to generalize to non-language modalities remains unproven.
3. The boundary conditions of RAJ (e.g., efficacy with extremely large expert counts) and the justification for choosing RAJ similarity over alternatives (e.g., cosine similarity) are not clarified or tested, limiting confidence in its robustness across diverse scenarios.
Questions

**Questions:**

1. MoE architectures are increasingly scaling to thousands of experts (e.g., larger Switch Transformer variants). Could you clarify: (1) Whether RAJ and GH losses maintain their effectiveness when the number of experts is drastically increased (e.g., 1k+ experts), where near-neighbor rank flips might become more frequent or impactful? (2) Does the Expert-Aware Scale (ES) incur additional computational overhead or require hyperparameter retuning when adapting to such large expert pools, and if so, how is this trade-off managed?
2. The paper proposes RAJ and GH losses to stabilize routing but provides no theoretical derivation explaining how these losses influence the core quantization parameters (e.g., scale and zero-point) in min-max or MSE-based PTQ. For example: (1) How do RAJ’s rank-alignment objectives and GH’s margin-preservation constraints modify the calculation of per-expert/per-channel scales in ES? (2) Can you provide mathematical proof or analysis showing that these losses minimize quantization error propagation in the router, rather than relying solely on empirical ablation results? This would strengthen the rationale for your method’s design beyond experimental observation

---

> ### Author Response · Authors · 2025-11-18
> **Response to Reviewer XAzX**
>
> Dear Reviewer XAzX,
>
> Thank you for your detailed review and for recognizing our analysis of router-induced error cascading and the complementary roles of ES, RAJ, and GH. We appreciate your comments regarding expert usage patterns, multimodal generalization, and the theoretical foundations of our loss designs. We address each of your concerns and questions below.
>
> ---
>
> > **W1**: The paper focuses heavily on optimizing router quantization (via RAJ and GH losses) but overlooks a core attribute of MoE architectures: variations in expert usage frequency and their impact on quantization performance. As the critique notes, MoE’s essence lies in dynamic expert utilization, yet the study provides no analysis of how differing expert usage patterns (e.g., frequently vs. rarely activated experts) affect quantization efficacy. It only addresses router-related errors and does not explore or mitigate disparities in quantization effects across individual experts—leaving a gap in its coverage of MoE-specific challenges and narrowing the method’s comprehensiveness.
>
> We agree that experts can exhibit different activation frequencies and behaviors, and studying expert-specific quantization effects is indeed an interesting direction. However, this aspect is orthogonal to the core focus of our work.
>
> Our goal is to isolate and address the router-induced quantization errors, which we show, through both analysis and ablations, account for a significant portion of the degradation that remains even after expert FFNs are quantized. This perspective is fundamentally different from EAQuant, which focuses on expert-weight quantization and does not examine routing stability or rank-flip behavior.
>
> Because our contribution centers on the routing mechanism and its sensitivity under quantization, we intentionally focus on router-side errors rather than expert-level variance. The latter is complementary and falls outside the scope of this study, but we agree it is a valuable direction that can build on top of our findings.
>
> > **W2**: The paper’s evaluation is limited to language-only MoE models (OLMoE, DeepSeek-MoE, Qwen3-MoE) and does not extend to multimodal MoE settings. Given that multimodal data heterogeneity could exacerbate routing instability, the method’s ability to generalize to non-language modalities remains unproven.
>
> Thank you for the comment. To address this concern and demonstrate that our framework generalizes beyond language-only MoE models, we additionally evaluate it on Qwen3-VL-30B-A3B-Instruct. We follow an analogous calibration setup using image–text pairs from Flickr30k and apply the same W4A4 quantization configuration as in our experiments. Evaluation is conducted with lmms-eval across a broad suite of multimodal benchmarks including GQA, ChartQA, ScienceQA, RealWorldQA, K-DTCBench, OCRBench, MMMU, and AI2D.
>
>
> | Method        | GQA   | ChartQA | ScienceQA | RWQA  | K-DTC | OCR   | MMMU  | ai2d  | Average |
> |---------------|-------|---------|-----------|-------|-------|-------|-------|-------|---------|
> | FP16      | 64.04 | 85.28 | 93.63 | 66.14 | 83.33 | 84.80 | 52.00 | 86.08 | 76.91 |
> | DuQuant       | 59.42 | 79.52 | 90.71 | 63.22 | 79.21 | 81.28 | 48.19 | 82.36 | 72.99 (-) |
> | MoEQuant      | 60.26 | 80.28 | 91.93 | 63.22 | 80.55 | 82.31 | 48.22 | **83.97** | 73.84 (1.17%) |
> | EAQuant       | 61.92 | 82.44 | 91.56 | 63.82 | 80.73 | 82.58 | 49.13 | 83.16 | 74.42 (1.96%) |
> | **ExpertQuant** | **62.08** | **84.17** | **92.17** | **64.08** | **81.19** | **82.98** | **50.90** | 83.16 | **75.09 (2.88%)** |
>
> Across all datasets, our method consistently outperforms existing baselines, with especially strong gains on multimodal reasoning and knowledge-intensive tasks. These results confirm that the proposed router-oriented quantization framework is not only effective for language MoE models but also transfers robustly to complex multimodal systems.
>
> We have added these experiments and the corresponding discussion as in Section 5.
>
> > **W3-1/Q1-1**: Whether RAJ and GH losses maintain their effectiveness when the number of experts is drastically increased (e.g., 1k+ experts), where near-neighbor rank flips might become more frequent or impactful?
>
>
> Thank you for the question. RAJ and GH operate **only on the top-k activated experts**, not on the full expert pool. Therefore, their effectiveness is determined by the stability of the *top-k margin*, and is fundamentally independent of whether the model has 64, 256, or 1,000+ total experts. Even in very large-MoE settings, only the top-k logits influence routing, so the number of non-selected experts does not affect our losses.
>
> While we are unable to run 1k-expert models due to resource constraints, our results already include SOTA models with **128 experts (e.g., Qwen3-MoE)** and show consistent improvements, supporting the scalability of our method.

---

> > ### Author Response · Authors · 2025-11-18
> > **Response to Reviewer XAzX**
> >
> > > **W3-2**: The justification for choosing RAJ similarity over alternatives (e.g., cosine similarity) are not clarified or tested, limiting confidence in its robustness across diverse scenarios.
> >
> > To evaluate the choice of RAJ similarity, we replaced RAJ with KL divergence and cosine similarity, computed strictly on the top-$k$ experts as shown in the below table.
> >
> > | Method        | ARC-C | ARC-E | BoolQ | HellaS | OBQA | RTE   | Wino. | Avg. ↑ |
> > |---------------|-------|-------|-------|--------|------|-------|-------|--------|
> > | ES+GH+KL      | 41.98 | 71.59 | **67.92** | 55.87 | 29.00 | 67.87 | **66.30** | 57.22 |
> > | ES+GH+Cosine  | 42.66 | 71.17 | 67.13 | 55.77 | 30.60 | 66.79 | 65.43 | 57.08 |
> > | ES+GH+RAJ     | **43.17** | **72.85** | 65.57 | **56.21** | **32.00** | **69.31** | 65.67 | **57.83** |
> >
> > While these alternatives align logits within the selected set, they are fundamentally magnitude- or angle-based and not inherently sensitive to ordering changes, which are the primary source of router errors under quantization. In contrast, RAJ explicitly compares the ordered top-$k$ sets with geometric rank weights and is invariant to affine logit transformations. This makes RAJ directly responsive to the near-neighbor rank flips that dominate MoE routing instability. As a result, KL and cosine provide weaker guidance for preserving routing consistency.
> >
> > We have added a new paragraph in Section 4.5 to discuss the choice of alignment.
> >
> > > **Q1-2**: Does the Expert-Aware Scale (ES) incur additional computational overhead or require hyperparameter retuning when adapting to such large expert pools, and if so, how is this trade-off managed?
> >
> >
> > Thank you for raising this concern. We would like to clarify that the runtime behavior of ES has been evaluated in Appendix C: *Runtime Efficiency*.
> >
> > As shown in Appendix C, ES introduces only negligible runtime overhead. The token/s measurements below demonstrate that ES does not incur extra overhead during inference and therefore requires no trade-off or additional tuning, even when the number of experts grows.
> >
> > | Model       | OLMoE       | DeepSeek     | Qwen3       |
> > | ----------- | ----------- | ------------ | ----------- |
> > | DuQuant     | 6.52 ± 0.01 | 23.88 ± 0.05 | 1.17 ± 0.01 |
> > | MoEQuant    | 6.84 ± 0.03 | 24.56 ± 0.06 | 1.21 ± 0.02 |
> > | EAQuant     | 6.72 ± 0.05 | 24.14 ± 0.07 | 1.20 ± 0.01 |
> > | ExpertQuant | 6.49 ± 0.03 | 23.79 ± 0.06 | 1.16 ± 0.01 |
> >
> >
> > > **Q2-1**: How do RAJ’s rank-alignment objectives and GH’s margin-preservation constraints modify the calculation of per-expert/per-channel scales in ES?
> >
> > RAJ and GH operate exclusively on the router logits and are designed to stabilize the routing decisions. They do not modify the quantization process of expert FFNs and therefore do not affect the calculation of per-expert or per-channel scales in ES. ES is applied independently to the expert weights, while RAJ and GH only adjust the router outputs to preserve top-k ranking under quantization.

---

> > > ### Author Response · Authors · 2025-11-18
> > > **Response to Reviewer XAzX**
> > >
> > > > **Q2-2**: Can you provide mathematical proof or analysis showing that these losses minimize quantization error propagation in the router, rather than relying solely on empirical ablation results? This would strengthen the rationale for your method’s design beyond experimental observation
> > >
> > > We thank the reviewer for pointing this out. RAJ and GH are router losses used during (post-training) optimization that shape the quantized router logits.
> > >
> > > Concretely, in the revised manuscript we now add a rank-aware MoE error decomposition (Appendix E.1, Lemma 1) for the theoretical analysis:
> > >
> > > $$
> > > | y^{(q)}(x) - y^{(\mathrm{fp})}(x) |*2
> > > \le C(x)\Big(2,d_{\text{rank}}(I,J) * \sum_{j\in I\cap J}|\pi_j^{(q)} - \pi_j^{(\mathrm{fp})}|
> > >   \Big),
> > > $$
> > >
> > > where $d_{\text{rank}}(I,J)$ is the number of rank mismatches between the FP and quantized top-k expert lists, and $\pi_j^{(\cdot)}$ are the router weights. This shows that routing error is determined by:
> > >
> > > 1. Which experts (and ranks) are selected (first term), and
> > > 2. How the routing weights differ when the same experts are selected (second term).
> > >
> > > We then prove two properties:
> > >
> > > * **RAJ as the rank disagreement:**
> > > RAJ is a rank-aware Jaccard on the ordered top-k lists. In Appendix E.2, we show that RAJ is equivalent (up to constants) to a rank-weighted disagreement $d_{\text{rank},w}(I,J) = \sum_r w_r \mathbf{1}[i_r \neq j_r]$, and thus directly upper-bounds $d_{\text{rank}}(I,J)$. Importantly, RAJ is invariant to affine transforms of the logits, so it does not depend on the absolute scale; it only pushes the quantized router to preserve the FP16 ordering. This explains how the optimization driven by RAJ steers the quantized logits to the full-precision configurations.
> > > * **GH as the weight disagreement:**
> > > GH is defined on differences of differences (margins between consecutive experts). We show in Appendix E.3 that the consecutive gaps $\Delta_r = r_r - r_{r+1}$ uniquely determine the logits up to an additive constant, and, via softmax smoothness, margin discrepancies $|\Delta_r^{(fp)} - \Delta_r^{(\mathrm{r})}|$ yield a bound:
> > > $$
> > > \sum_{j\in I\cap J}|\pi_j^{(fp)} - \pi_j^{(\mathrm{r})}|
> > > \le \text{const} \cdot L_{\mathrm{GH}}.
> > > $$
> > > Thus GH explicitly controls the **second term** in the MoE error bound by keeping quantized margins close to the FP margins, which in turn stabilizes the routing weights for any MSE-optimal scale.
> > >
> > > Putting these together, the new analysis clarifies the role of RAJ and GH precisely by controlling the two terms in the bound above.
> > >
> > > ---
> > >
> > > Thank you once again for your valuable feedback, which has significantly improved our work. We welcome further questions or suggestions for refinement.

---

> > > > ### Comment · Reviewer_XAzX · 2025-11-25
> > > >
> > > > Thank you for your detailed response. However, I still tend to maintain the original score.
> > > >
> > > > The core of the research lies in the errors caused by the quantization of the router. Yet the authors mention: "...a significant portion of the degradation that remains even after expert FFNs are quantized." This suggests that most of the error stems from the outputs of the FFNs. If the router is indeed as important as the authors claim, full precision could be adopted—after all, both the parameter count and computational complexity of the router are negligible in the entire MoE architecture.
> > > >
> > > > Methodologically, the authors designed two losses for the router to optimize expert selection, which essentially cannot reduce the errors arising from computations during expert quantization. Experimentally, the ablation studies focus on comparisons of expert selection, leaving it unclear whether the quantization error of the actual MoE outputs has been reduced after router optimization. The presentation seems to avoid addressing the key issue and emphasize less critical aspects, which strikes me as odd overall.
> > > >
> > > > In summary, I believe this paper falls below the acceptance threshold.

---

> > > > > ### Author Response · Authors · 2025-11-25
> > > > > **Response to Reviewer XAzX**
> > > > >
> > > > > Thank you for the thoughtful follow-up.
> > > > >
> > > > > ---
> > > > >
> > > > > We fully agree that keeping the router in full precision is a valid and widely used design choice. At the same time, our work focuses on the complementary scenario: when the router is quantized, how can we mitigate the resulting accuracy degradation? Prior MoE quantization studies that serve as our baselines also evaluate models under this setting. Thus, our method does not conflict with FP-router approaches; rather, it offers a practical solution for situations in which router quantization is necessary. Since both design choices appear in real-world deployments, studying router quantization remains important and cannot be overlooked.
> > > > >
> > > > > Regarding quantization error, our experiments across multiple tasks show that addressing router-induced discrepancies leads to consistent and measurable accuracy improvements. These findings further support the motivation and utility of the proposed method.
> > > > >
> > > > > Lastly, we would like to emphasize that research advances through the exploration of diverse design options. Both FP-router and quantized-router configurations are meaningful depending on deployment constraints, and we believe that examining the quantized-router setting is both necessary and beneficial for strengthening MoE quantization as a whole.

---

### Official Review · Reviewer_5sFX · 2025-10-31

**Soundness:** 2
**Presentation:** 2
**Contribution:** 2
**Rating:** 4
**Confidence:** 3

**Summary:**

The paper proposes that router ranking instability caused by quantization is the main reason for the performance degradation of MoE. It combines expert-aware scaling and rank-aware loss to achieve training-independent MoE quantization. This significantly reduces quantization error while maintaining high inference accuracy across several mainstream MoE models.

**Strengths:**

1. The paper identifies and systematically analyzes the router sensitivity problem in MoE quantization.
2. Discrepancies in input distribution among experts were identified, and Expert-Aware Scale (ES) was proposed to address this issue.
3. All calibration targets (ES, RAJ, GH) can be completed in the post-training phase without backpropagation or additional fine-tuning.

**Weaknesses:**

1. The experimental design is incomplete, and the core arguments are not sufficiently verified. The paper claims that "router decision-making is a key factor in the quantization degradation of MoE," but the experiment in Figure 1 only shows a partial comparison of q/k/v/router, without clearly stating whether shared and non-shared experts are quantified, and the experimental section also does not fully explain.
2. The performance gain of routers is limited, and the conclusion is overstated. As shown in Figure 1, maintaining full precision in the router only brings about a +1% improvement in accuracy, while expert quantization errors typically cause a 5–10% decrease.
3. Lack of comprehensive quantization and interactive ablation analysis: In actual deployments, when both the router and experts perform quantization simultaneously, the benefits of improved router consistency are often overshadowed by expert errors.
4. Expert-Aware Scale (ES) increases complexity and raises questions about its engineering feasibility. ES introduces an independent scale for each expert and each channel. Although mathematically equivalent to reparameterization, in practice, additional parameters need to be stored, and the inference path involves an extra activation scaling. It cannot be integrated into the static weight graph like AWQ. The paper does not discuss these additional costs, nor does it provide an assessment of the inference overhead.

**Questions:**

Please see weaknesses.

---

> ### Author Response · Authors · 2025-11-18
> **Response to Reviewer 5sFX**
>
> Dear Reviewer 5sFX,
>
> Thank you for your review and for highlighting our analysis of router sensitivity and the advantages of calibration-only quantization. We appreciate your comments regarding the experimental design, the magnitude of router gains, and the engineering considerations of ES. We address each of these concerns in detail below.
>
> ---
>
> > **W1**: The experimental design is incomplete, and the core arguments are not sufficiently verified. The paper claims that "router decision-making is a key factor in the quantization degradation of MoE," but the experiment in Figure 1 only shows a partial comparison of q/k/v/router, without clearly stating whether shared and non-shared experts are quantified, and the experimental section also does not fully explain.
>
>
> Thank you for the helpful comment. We apologize for the lack of clarity in the experimental description.
>
> **Expert quantization.** We confirm that all experts—both shared and non-shared—are quantized in every experiment throughout the paper, including Figure 1. This reflects realistic deployment constraints: in typical MoE architectures, expert FFNs alone account for over 95% of total parameters, so leaving them in FP32 would eliminate most memory and latency benefits of quantization. To avoid ambiguity, we will explicitly state this in the revision. Importantly, in the main experiment section, all linear layers across the model (attention, MLP, router, and expert FFNs) are quantized under the stated bit-width settings.
>
> **Purpose of Figure 1.** The purpose of Figure 1 is *not* to show that router-only quantization dominates total accuracy loss, but rather to **isolate** the router’s sensitivity by holding *expert quantization constant* and comparing it against other non-expert components (q/k/v, attention projections, gating, etc.). By quantizing experts in all cases, we ensure a fair comparison and highlight that:
>
> * Once experts are quantized, the **remaining drop** in MoE performance is primarily driven by router instability, not by q/k/v or other modules.
> * Even small perturbations in router logits cause **top-k rank flips**, which cascade into incorrect expert selection and amplify downstream errors.
>
> We have updated the paper to clarify these settings in Section 1.
>
> > **W2**: The performance gain of routers is limited, and the conclusion is overstated. As shown in Figure 1, maintaining full precision in the router only brings about a +1% improvement in accuracy, while expert quantization errors typically cause a 5–10% decrease.
>
> We respectfully clarify that our conclusion is not overstated, and we never claim that router quantization alone accounts for the entire degradation. Rather, Figure 1 is designed to isolate the marginal impact of each non-expert module once expert FFNs are already quantized. Under this controlled setup, keeping only the router in FP32 yields a ~+1% gain, an effect that may appear small in absolute terms but is highly consequential, because even tiny perturbations in router logits trigger top-k rank flips, which in turn reshape expert usage and propagate errors across the entire MoE computation path.
>
> Our contribution is not the magnitude of improvement alone, but the identification of router-induced rank instability as a previously overlooked source of error. To our knowledge, this is the first work to systematically analyze and quantify the role of the router in MoE PTQ, and also the first to propose a dedicated solution targeting router consistency, demonstrating that stabilizing routing recovers the post-quantization loss.
>
> > **W3**: Lack of comprehensive quantization and interactive ablation analysis: In actual deployments, when both the router and experts perform quantization simultaneously, the benefits of improved router consistency are often overshadowed by expert errors.
>
> We have already conducted an ablation specifically designed to analyze this interaction.
>
> In our "perfect matching" experiment (Figure 8), we keep experts quantized (W4A4) and substitute only the router’s top-k decisions with the FP16 ones. On OLMoE, the full FP16 model achieves an average score of 60.64. Under W4A4, a baseline PTQ method such as DuQuant reaches 55.35, but simply correcting router mismatches, without modifying expert quantization at all, raises the score to 57.35.
>
> This +2.0 improvement is substantial given that:
>
> * the router accounts for only 0.03% of parameters, yet
> * its errors can cascade through expert selection and dominate the remaining gap once expert quantization is fixed.
>
> Thus, the results show that router consistency is not overshadowed by expert errors; rather, it becomes a critical residual bottleneck after expert quantization is applied. Our ablation directly validates this interaction and motivates the need for router-aware design in MoE quantization.

---

> > ### Author Response · Authors · 2025-11-18
> > **Response to Reviewer 5sFX**
> >
> > > **W4**: Expert-Aware Scale (ES) increases complexity and raises questions about its engineering feasibility. ES introduces an independent scale for each expert and each channel. Although mathematically equivalent to reparameterization, in practice, additional parameters need to be stored, and the inference path involves an extra activation scaling. It cannot be integrated into the static weight graph like AWQ. The paper does not discuss these additional costs, nor does it provide an assessment of the inference overhead.
> >
> > Thank you for raising this concern. We would like to clarify that the engineering feasibility of Expert-Aware Scale (ES) has been thoroughly evaluated in Appendix C: *Runtime Efficiency*.
> >
> > As shown in Appendix C, ES introduces only negligible runtime overhead. We also provide the table below (token/s) demonstrating that ES does not incur the extra activation scaling the reviewer is concerned about during inference.
> >
> > | Model       | OLMoE       | DeepSeek     | Qwen3       |
> > | ----------- | ----------- | ------------ | ----------- |
> > | DuQuant     | 6.52 ± 0.01 | 23.88 ± 0.05 | 1.17 ± 0.01 |
> > | MoEQuant    | 6.84 ± 0.03 | 24.56 ± 0.06 | 1.21 ± 0.02 |
> > | EAQuant     | 6.72 ± 0.05 | 24.14 ± 0.07 | 1.20 ± 0.01 |
> > | ExpertQuant | 6.49 ± 0.03 | 23.79 ± 0.06 | 1.16 ± 0.01 |
> >
> > ---
> >
> > Thank you once again for your valuable feedback, which has significantly improved our work. We welcome further questions or suggestions for refinement.

---

### Note · Authors · 2025-12-03

I have read and agree with the venue's withdrawal policy on behalf of myself and my co-authors.